# Comprehensive analysis of genes associated with necroptosis and pyroptosis in intestinal ischemia-reperfusion injury

**Shuang Bao**[1][☺], **YanBo Sun**[1][☺], **YiChen Hu**[1][☺], **XueFen Lei**[2][☺], **YuanPei Zhao**[1], **WeiMing Li**[1]*

**1** Department of Gastrointestinal Surgery, The Second Affiliated Hospital of Kunming Medical University, Kunming, Yunnan, China, **2** Department of Medical Oncology, The Second Affiliated Hospital of Kunming Medical University, Kunming, Yunnan, China

☺ These authors contributed equally to this work.
* liweiming@kmmu.edu.cn

## Abstract

### Background

Intestinal ischemia–reperfusion (II/R) injury is a severe clinical condition in which regulated cell death programs—including pyroptosis and necroptosis—have emerged as key drivers of tissue damage and inflammation. We sought to delineate cell-death–related molecular signatures and candidate therapeutic targets in II/R injury.

### Methods

We obtained transcriptome datasets from Gene Expression Omnibus (GEO) databases for mice (GSE96733, GSE232246) and humans (GSE37013). We cross-referenced genes associated with necroptosis and pyroptosis with differentially expressed genes to identify death-related features. Hub genes were identified through the topological structure of protein interaction networks and validated using an internal validation set, an independent validation set, WGCNA, and qRT-PCR. These genes were also associated with immune cell infiltration. Drug–gene interactions were predicted using DGIdb and verified through molecular docking.

### Results

We identified 1,027 differentially expressed genes (DEGs) in the training set and derived 7 cell death-related differentially expressed genes (DCDEGs) by intersecting gene sets associated with necroptosis and pyroptosis. PPI-based prioritization identified four hub genes—*Il1β, Ripk3, Sting1 (Tmem173),* and *Tnfaip3*—suggesting cross-regulatory interactions between inflammation and cell death in ischemia-reperfusion pathology. These hub genes were validated using WGCNA analysis and an internal validation set. Immune infiltration analysis indicated significant correlations between hub genes and multiple immune compartments. A predictive model

**Data availability statement:** All relevant data are within the manuscript and its Supporting information files.

**Funding:** The author(s) declare that financial support was received for the research and/or publication of this article. This work was funded by the National Natural Science Foundation of China (NSFC) (No. 82460114) and foreign cooperative research project of the Second Affiliated Hospital of Kunming Medical University (No.2022dwhz09). The funders had no role in study design, data collection and analysis, decision to publish, or preparation of the manuscript.

**Competing interests:** The authors declare that the research was conducted in the absence of any commercial or financial relationships that could be construed as a potential conflict of interest.

showed good discrimination in the discovery data, and 54 candidate drugs targeting the hub genes were retrieved. qRT-PCR confirmed dysregulation of three hub genes.

## Conclusion

*Il1β, Ripk3, Sting1*, and *Tnaip3* were identified as hub genes associated with necroptosis and pyroptosis in intestinal ischemia-reperfusion (II/R) injury. This study provides a reproducible framework and identifies testable targets for translational exploration.

## Introduction

Intestinal ischemia-reperfusion (II/R) injury is a prevalent pathological process with high mortality rates in clinical practice, frequently observed in scenarios such as abdominal aortic aneurysm surgery, mesenteric artery embolism, traumatic shock, and organ transplantation [1]. It often leads to severe inflammatory responses and tissue damage [2].

The main process of II/R injury involves a vicious cycle of oxidative stress and inflammation. During the ischemic phase, tissue hypoxia causes the conversion of xanthine dehydrogenase to xanthine oxidase, leading to the accumulation of hypoxanthine. Subsequently, during the reperfusion phase, the reintroduction of oxygen triggers the explosive generation of reactive oxygen species (ROS), such as superoxide anion and hydrogen peroxide [3]. These ROS not only directly damage DNA, proteins, and lipids, but also activate the NF-κB signaling pathway, thereby promoting the release of pro-inflammatory factors such as tumor necrosis factor-α (*TNF-α*) and interleukin-1β (*IL1β*) [4]. Furthermore, ROS triggers endothelial cell activation, upregulates the expression of adhesion molecules (such as *ICAM-1* and *VCAM-1*), and recruits neutrophil infiltration. Activated neutrophils exacerbate tissue damage by releasing proteases and myeloperoxidase, while also promoting the release of platelet-activating factor (PAF), thereby worsening microcirculatory dysfunction [5]. This series of reactions leads to disruption of the intestinal barrier function. Due to the high oxygen consumption rate of epithelial cells at the tips of intestinal villi, they are highly susceptible to necrosis during ischemia-reperfusion, which in turn causes degradation of tight junction proteins (such as ZO-1 and Claudin-2). This ultimately results in bacterial translocation and endotoxin release, activating a systemic inflammatory response [6,7]. Due to its high metabolic activity and unique vascular network structure, the intestine is particularly susceptible to ischemia-reperfusion injury (IRI). Following injury, the intestine is prone to barrier dysfunction, bacterial translocation, and systemic inflammatory response syndrome (SIRS), which may ultimately progress to multiple organ failure (MOF) [2].

In recent years, the roles of necroptosis and pyroptosis—two major forms of programmed cell death—in driving intestinal ischemia-reperfusion injury have become increasingly apparent. Both mechanisms exacerbate injury through proinflammatory signaling pathways and have emerged as novel therapeutic targets [5,8].

Necroptosis is a form of necrotic cell death under regulatory control that depends on the *RIPK1-RIPK3-MLKL* signaling pathway. Triggering factors such as *TNF-α*, *IFN-γ*, and LPS activate *RIPK1* through death receptors (*TNFR1*) or pattern recognition receptors (*TLR3/4*) [9,10]. In II/R injury, *TNF-α* released during the ischemic phase binds to *TNFR1* to form complex I (containing *TRADD* and *RIPK1*), which is deubiquitinated by *CYLD* to form the pro-apoptotic complex IIb (necrosome) [9]. During the execution phase, *RIPK3* phosphorylates *MLKL*, inducing its oligomerization and translocation to the cell membrane, where it forms ion channels leading to cell swelling and rupture. The research clearly showed that in the case of II/R injury, the lack of *RIPK3* specifically in endothelial cells results in heightened vascular permeability, increased levels of the inflammatory cytokine *IL-6*, and enhanced expression of *MLKL*. Furthermore, the absence of *RIPK3* facilitates the upregulation of endothelial adhesion molecules (such as *VCAM-1* and *ICAM-1*), which worsens leukocyte infiltration in both the small intestine and lungs, and is directly linked to the pathological mechanisms underlying intestinal barrier damage [11]. Furthermore, beyond membrane disruption, oligomerized *MLKL* can also activate the *PGAM5*/*DRP1* pathway, inducing excessive mitochondrial fission and accelerating cellular demise [12]. The rupture of necroptotic cells releases damage-associated molecular patterns (DAMPs) such as *HMGB1* and ATP, which recruit macrophages and further amplify the inflammatory cascade, directly compromising the intestinal epithelial barrier [12].

Pyroptosis, a type of programmed cell death marked by cellular breakdown, is primarily driven by the inflammasome-caspase-gasdermin D (*GSDMD*) pathway, encompassing both classical and non-classical routes. In the classic pathway, DAMPs and ROS generated during II/R activate the *NLRP3* inflammasome, which recruits and activates *Caspase-1*. Active *Caspase-1* then cleaves *GSDMD*, liberating its N-terminal fragment (*GSDMD-p30*) [13]. This fragment oligomerizes to form pores in the plasma membrane, causing cytokine release (*IL1β* and *IL-18*) and lytic cell death [14]. Alternatively, in the non-canonical pathway, cytoplasmic LPS from translocated bacteria can directly activate human *Caspase-4/5* or murine *Caspase-11*, which also cleave *GSDMD* to execute pyroptosis while concurrently activating the *NLRP3* inflammasome to amplify the inflammatory response [15]. In the context of II/R, pyroptosis in intestinal epithelial cells creates a vicious cycle: *GSDMD* pores accelerate $K^+$ efflux, which further activates *NLRP3*, forming a positive feedback loop that intensifies the local "inflammatory storm." Moreover, *IL-1β* released during pyroptosis can mediate remote organ damage (e.g., acute lung injury) by promoting neutrophil infiltration and increasing microvascular permeability [16].

In summary, necroptosis and pyroptosis function as synergistic amplifiers of inflammation in II/R injury, critically contributing to both local tissue destruction and the systemic inflammatory response [17]. Targeting key molecular nodes within these pathways or their crosstalk holds promise for overcoming the limitations of conventional antioxidant and anti-inflammatory therapies. To this end, this study conducted a comprehensive bioinformatic analysis of differentially expressed mRNAs associated with necroptosis and pyroptosis following intestinal IRI. Our aim is to systematically elucidate the expression profiles, functional networks, and potential therapeutic targets of these genes, thereby providing novel insights and strategic directions for the treatment of II/R injury.

## Materials and methods

### Data sources and study design

The raw transcriptomic data for mouse datasets GSE96733 and GSE232246, along with human dataset GSE37013, were downloaded from the GEO database (https://www.ncbi.nlm.nih.gov/geo/). We designated the GSE96733 dataset, based on the GPL23038 [Clariom_S_Mouse] platform, as the primary cohort and partitioned it into a training set (4 control groups and 4 II/R 3 h samples) and an internal validation set (4 control groups and 4 II/R 6 h samples). Additionally, the GSE232246 dataset (3 controls and 3 II/R samples) based on the GPL21103 [Illumina HiSeq 4000] platform serves as an independent validation set. For cross-species validation, we additionally selected 21 samples (7 controls and 14 II/R injury samples) from the human intestinal ischemia-reperfusion injury dataset GSE37013 as an external validation set. This dataset was generated on the Illumina HumanHT-12 V3.0 platform (GPL6947). All datasets were processed using standard bioinformatics workflows. Missing values were imputed, negative values were normalized to zero, and duplicate

genes were averaged. The resulting gene expression matrix was used for subsequent analyses. Concurrently, we systematically compiled mouse gene sets associated with necroptosis and pyroptosis from the NCBI Gene database (https://www.ncbi.nlm.nih.gov/gene). Specifically, using its advanced search function, we searched for "necroptosis" [All Fields] and 'pyroptosis' [All Fields] respectively, limiting the species to "Mus musculus." Through manual review of the preliminary results, we retained only genes with clear functional annotations and names, ultimately identifying 168 necroptosis-related genes and 299 pyroptosis-related genes for subsequent analysis. The research procedure is depicted in Fig 1. The complete code for this analysis, see S1 Table.

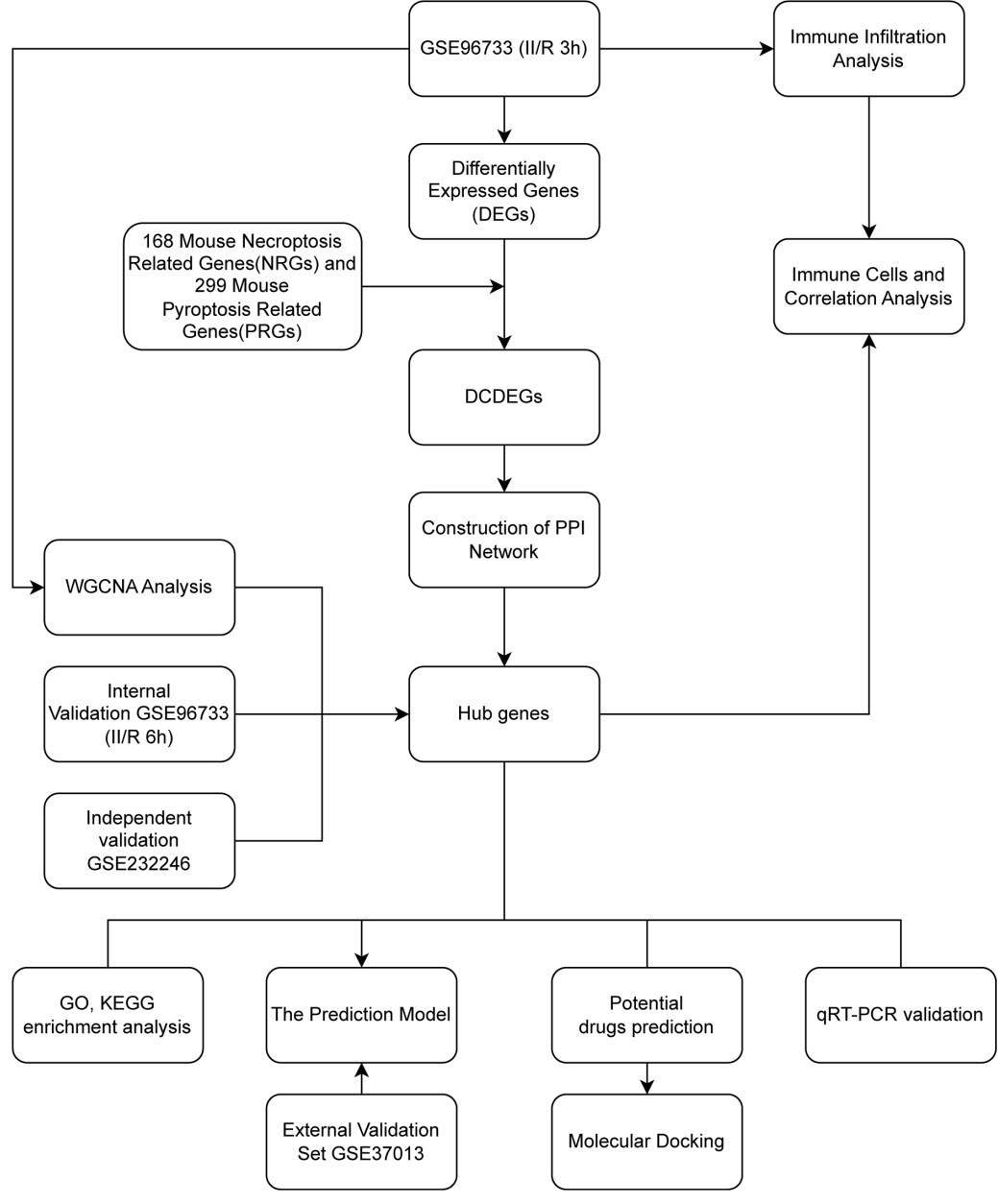

**Fig 1. Experimental design roadmap.**

## Differential expression analysis

To identify core transcriptional changes in intestinal ischemia-reperfusion injury (IRI), we performed differential expression gene (DEG) analysis between control and IRI groups in the training dataset (GSE96733). This analysis was conducted using the "limma" package (version 3.64.1) in R software (version 4.5.0). DEGs were filtered using a threshold of $|\log_2 FC| > 1$ and an adjusted $p$-value $< 0.05$. Volcano plots and heatmaps were generated using the "ggplot2" package (version 3.5.2) to visualize the differential expression results.

## Selection of cell death-related genes

Cell death can be categorized into several major forms, including apoptosis, necroptosis, autophagy, ferroptosis, pyroptosis, and necrosis, each characterized by distinct morphological and pathophysiological features. This study focuses on two of these forms—necroptosis and pyroptosis—and their roles in intestinal ischemia-reperfusion (II/R) injury. We first obtained mouse genes associated with these processes from the NCBI database, identifying 168 necroptosis-related genes (NRGs) and 299 pyroptosis-related genes (PRGs). Subsequently, the differentially expressed genes (DEGs) from the GSE96733 (II/R 3 h) dataset were intersected with the NRGs and PRGs to derive necroptosis-related DEGs (DENRGs) and pyroptosis-related DEGs (DEPRGs). Finally, the DENRGs and DEPRGs were further intersected to identify a core set of cell death-related differentially expressed genes (DCDEGs). The expression profiles of these DCDEGs were visualized using a violin plot generated with the R package "ggplot2" (version 3.5.2).

## Protein-protein interaction network (PPI) construction and hub genes screening

A protein-protein interaction (PPI) network for proteins encoded by the cell death-related differentially expressed genes (DCDEGs) was constructed using the STRING database (version 12.0; https://cn.string-db.org/), with a combined confidence score threshold of > 0.4. The resulting network was imported into Cytoscape (version 3.10.3) for visualization and further analysis. Hub genes related to necroptosis and pyroptosis were identified using the CytoHubba plugin (version 0.1), which applied four distinct algorithms: MCC, MNC, Degree, and EPC. The MCC algorithm identifies hub nodes by considering their participation in maximal clusters within the PPI network, thereby enhancing sensitivity to key proteins [18]. MNC evaluates the local neighborhood connectivity of a node, reflecting its influence within a subnet [18]. Degree counts the direct connections of a node; while computationally efficient, it may favor highly connected nodes and overlook those with fewer but critical links [19]. EPC is a global topological method based on edge percolation theory, assessing node importance by simulating random edge failures across the network [20]. The top five genes from each algorithm were selected, and their overlap was determined to define a core set of hub genes associated with necroptosis and pyroptosis. Differential expression of these hub genes was visualized using a heatmap generated with the "pheatmap" R package (version 1.0.13).

## Validate hub gene expression in the internal validation set and independent validation set

To validate the expression levels of hub genes, we first compared the expression differences between the control group and the II/R group in the internal validation set (GSE96733, II/R 6 h) and performed statistical analysis using the non-parametric rank-sum test. To further enhance the reliability of the results, we employed an independent validation dataset (GSE232246). In addition to the rank-sum test, we supplemented the analysis with t-tests and calculated Cohen's d values to assess the effect size of expression differences between groups. All statistical analyses were performed using R software, and violin plots and boxplots were visualized using the "ggplot2" package (version 3.5.2).

## WGCNA analysis validates hub genes

Weighted Gene Co-expression Network Analysis (WGCNA) was performed on the GSE96733 (II/R 3 h) dataset using the "WGCNA" R package (version 1.73). The top 25% of genes with the highest variance were selected for network

construction to identify key modules significantly associated with the traits. The presence of the four hub genes within these key modules was subsequently verified, and the overlap was visualized using the "VennDiagram" package (version 1.7.3).

## GO and KEGG enrichment analysis of hub genes

Functional enrichment analysis of Gene Ontology (GO) terms (Biological Process, Cellular Component, Molecular Function) and Kyoto Encyclopedia of Genes and Genomes (KEGG) pathways was performed using the R package "clusterProfiler" (v4.16.0) [21]. Significantly enriched terms were screened with a significance threshold of $P < 0.05$. The results were visualized using the "ggplot2" (v3.5.2) and "enrichplot" (v1.28.2) packages within the R environment.

## Immune infiltration analysis

The landscape of immune cell infiltration was deconvoluted using the CIBERSORT algorithm. Subsequently, Spearman's rank-order correlation analysis was applied to assess the associations between the identified hub genes and the abundances of individual immune cell types [22]. These correlations, which delineate the relationship between necroptosis/pyroptosis and the immune microenvironment, were visualized using the "ggcorrplot" R package (version 0.1.4.1) [23].

## Predictive model building and validation

To evaluate the predictive efficacy of hub genes for intestinal ischemia-reperfusion (II/R) injury, we constructed a classification model using strongly regularized Lasso regression ($\lambda = 0.1$) based on expression data of four hub genes from a mouse training set (GSE96733, 4 Control vs 4 IRI) to mitigate the risk of overfitting in small samples. Subsequently, through orthologous gene conversion, we applied this model to an external validation dataset (GSE37013, 7 Control vs 14 IRI) and systematically evaluated its discriminative performance across species using ROC curves and AUC values [24].

## Drug–gene interaction mining

Known drug-gene interactions for the human homologs of the four hub genes were retrieved from the DGIdb database (https://dgidb.org/). The resulting interaction network was subsequently visualized using the "ggplot2" (version 3.5.2) and "visNetwork" (version 2.1.2) packages in R.

## Molecular docking validation

Molecular docking was performed to validate the predicted interactions and assess the binding affinities between the top candidate drugs (prioritized from the DGIdb database) and the proteins encoded by the hub genes. The three-dimensional structures of the target proteins were obtained by querying their amino acid sequences (retrieved from the UniProt Knowledgebase) against the SWISS-MODEL server for homology modeling. Protein and ligand structures were prepared for docking by removing water molecules, adding hydrogen atoms, and assigning partial charges using AutoDock Tools. Docking simulations were carried out with AutoDock Vina, where the search space was defined as a grid box encompassing the entire protein surface to allow for blind docking. For each ligand, 50 independent docking runs were conducted to extensively sample the conformational space. The resulting pose with the most favorable binding affinity (lowest binding energy) and highest conformational cluster population was selected as the representative binding mode. Visualization and analysis of the docking results were conducted using PyMOL and Discovery Studio.

## Gene expression analysis via quantitative RT-PCR

To validate the expression of hub genes identified through bioinformatics analysis, we employed qRT-PCR technology to quantitatively analyze the expression of Ripk3, Sting1, Tnfaip3, and IL1b in the small intestinal tissues of sham-operated

mice and mice with intestinal ischemia-reperfusion injury. The experiment utilized six male mice, with three samples in the sham-operated group and three in the model group. Prior to establishing the mouse intestinal I/R injury model, all six mice underwent 12-hour fasting without water restriction. Mice were anesthetized via intraperitoneal injection of pentobarbital. The superior mesenteric artery was occluded using a non-invasive micro-arterial clamp. The micro-arterial clamp was removed at 45 min. Mice were euthanized 3 hours after reperfusion, and small intestinal tissue was rapidly excised, immediately placed in liquid nitrogen for rapid freezing, and subsequently stored at −80°C. Total RNA was extracted using Trizol Reagent (Product No.: 15596026). Subsequently, total RNA was reverse transcribed into cDNA using the FastKing RT Kit (With gDNase) FastKing cDNA First Chain Synthesis Reagent Kit (Product No.: KR116). qPCR reactions were performed using Taq Pro Universal SYBR qPCR Master Mix qRT-PCR on the LightCycler 96 system. The reaction system comprised: 5 µL SYBR Green Master Mix, 0.25 µL forward primer, 0.25 µL reverse primer, 1 µL cDNA template, supplemented with nuclease-free water to a total volume of 10 µL. The reaction program was as follows: 95°C pre-denaturation for 30 seconds; followed by 40 cycles of 95°C for 10 seconds and 60°C for 30 seconds; followed by a melting curve analysis (95°C for 15 seconds, 60°C for 60 seconds, 95°C for 15 seconds). Each sample included three technical replicates, and the experiment was independently repeated three times (n = 3). β-actin was used as the internal reference gene, and gene expression levels were calculated using the $2^{-\triangle\triangle Ct}$ method; primer sequences are listed in Table 1. The protocol for this animal study received approval from the Experimental Animal Ethics Committee of Yunnan Labreal Biotechnology Co., Ltd (Case Number: SL20250202).

## Statistical analysis

Statistical analyses were performed using R software (version 4.5.0). Differential expression was defined as $|\log_2 FC| > 1$ with FDR-adjusted $p < 0.05$. Group comparisons used t-tests and Wilcoxon tests, with $p < 0.05$ considered significant. Correlations were assessed using Spearman's method, and predictive performance was evaluated by ROC-AUC analysis. A $p$-value $< 0.05$ was considered statistically significant.

## Results

### DEGs analysis

Differential expression analysis of the training set (GSE96733 3 h), conducted using the "limma" R package, identified 1,027 differentially expressed genes (DEGs). This set consisted of 605 up-regulated and 422 down-regulated transcripts (Fig 2A and 2B). For detailed information, see S2 Table.

### Identification of cell death-related genes

To identify genes associated with necroptosis and pyroptosis in II/R injury, we first curated 168 murine necroptosis-related genes (NRGs) and 299 pyroptosis-related genes (PRGs) from the NCBI Gene database. Intersecting these with the differentially expressed genes (DEGs) yielded 22 necroptosis-related DEGs (DENRGs) (Fig 3A) and 30 pyroptosis-related DEGs (DEPRGs) (Fig 3B). Finally, by overlapping DENRGs and DEPRGs, we screened out 7 DCDEGs (Fig 3C). Venn diagram analysis revealed that the 7 overlapping genes constitute the DCDEGs, including *S100a9, Il1β, Ripk3, Tnfaip3,*

**Table 1. qT-PCR amplification primers and their sequences.**

| Gene | Forward primer (5′–3′) | Reverse primer (3′–5′) |
|---|---|---|
| Ripk3 | GGCTCTCGTCTTCAACAA | GTAGTTCTTGGTGGTGCTA |
| Il1β | CAATGGACAGAATATCAAC | ACAGGACAGGTATAGATT |
| Sting1 | CACTGTATGGCTATGATTC | GACTTATAGAGGACCAGAA |
| Tnfaip3 | CGCTGTTCCACTTGTTAA | TTCCTTCATCTCATTCTCAG |

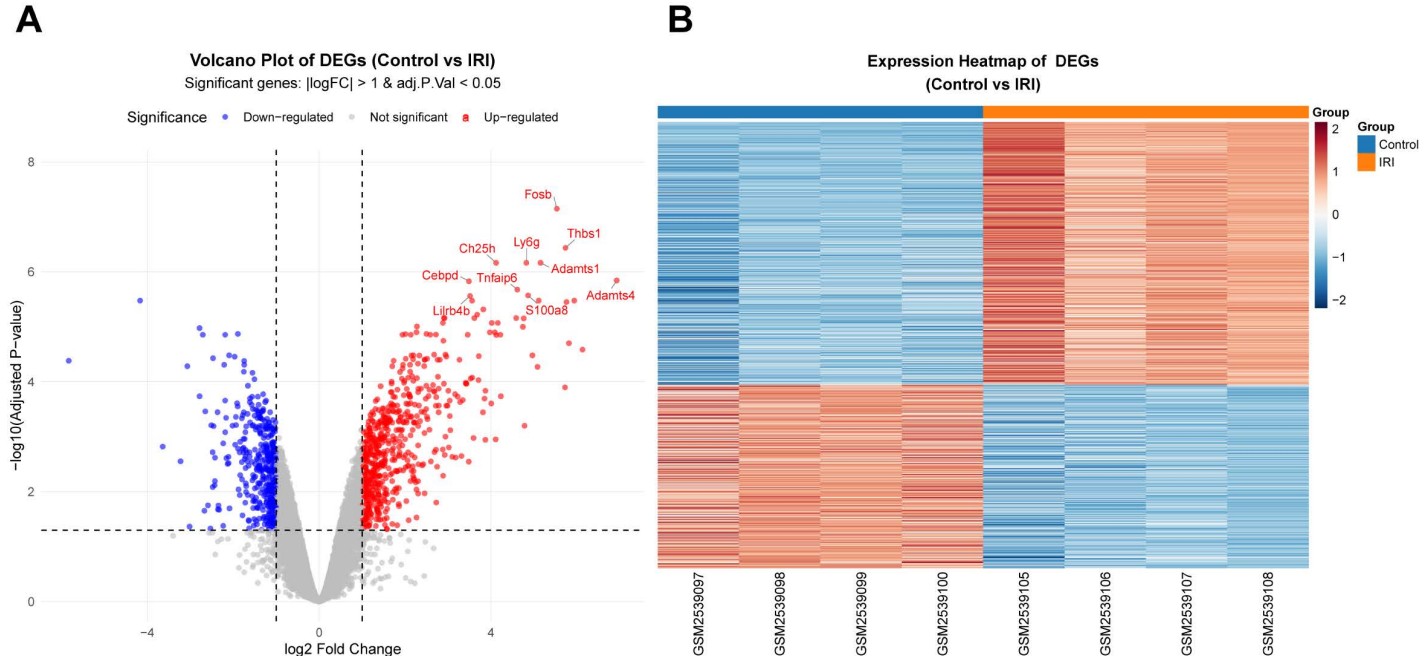

**Fig 2. Detection of DEGs. (A)** A volcano plot displays the genes that are expressed differently between the Control samples and the II/R injury samples in GSE96733 (II/R 3 h). Up-regulated genes are indicated by red dots, whereas down-regulated genes are shown with blue dots. The criteria established were |log₂FC|>1 and an adjusted $P$ value<0.05. **(B)** A heatmap illustrates the genes with differential expression between the Control samples and the II/R injury samples in GSE96733 (II/R 3 h). The color red represents DEGs that are significantly up-regulated in the samples, whreas blue signifies DEGs that are significantly down-regulated.

*Sting1, Hif1a, Tnc*. The expression levels of DCDEGs in Control and II/R-damaged group were visualized using the "ggplot2" package (Fig 3D). For detailed information, see S3 Table.

## Hub genes screening

We constructed a protein-protein interaction (PPI) network for the 7 cell death-related differentially expressed genes (DCDEGs) using the STRING database (Fig 4A). To identify pivotal hub genes mediating necroptosis-pyroptosis cross-talk, we then analyzed this network using the CytoHubba plugin with four topological algorithms (MCC, MNC, EPC, and Degree) (Fig 4B–4E). The four hub genes were determined by overlapping the top five genes from each of the four algorithms (Fig 4F). These hub genes include *Il1β, Ripk3, Tmem173 (Sting1), Tnfaip3*. A graphical representation of hub gene expression, contrasting control and II/R injury conditions, was produced utilizing the "pheatmap" package (Fig 4G). For detailed information, see S4 Table.

## Internal validation set and independent validation set for verifying hub genes

In the internal validation set (GSE96733 6 h), the transcriptional levels of the four hub genes were consistently reproduced with the analysis results from the training set (II/R 3 h) (Fig 5A). In the independent validation set (GSE232246), although Wilcoxon tests failed to reach statistical significance (all $p>0.05$) due to small sample sizes (n=3/group), all genes showed consistent upregulation trends (FC>1). Effect size (Cohen's d) analysis further indicated biological significance, with Il1b, Ripk3, and Sting1 exhibiting large effect sizes (d>0.8) and Tnfaip3 showing a moderate effect size (d>0.5). These findings suggest that the potential role of these pivotal genes in IRI is partially supported by the data (Fig 5B). For detailed information, see S5 Table.

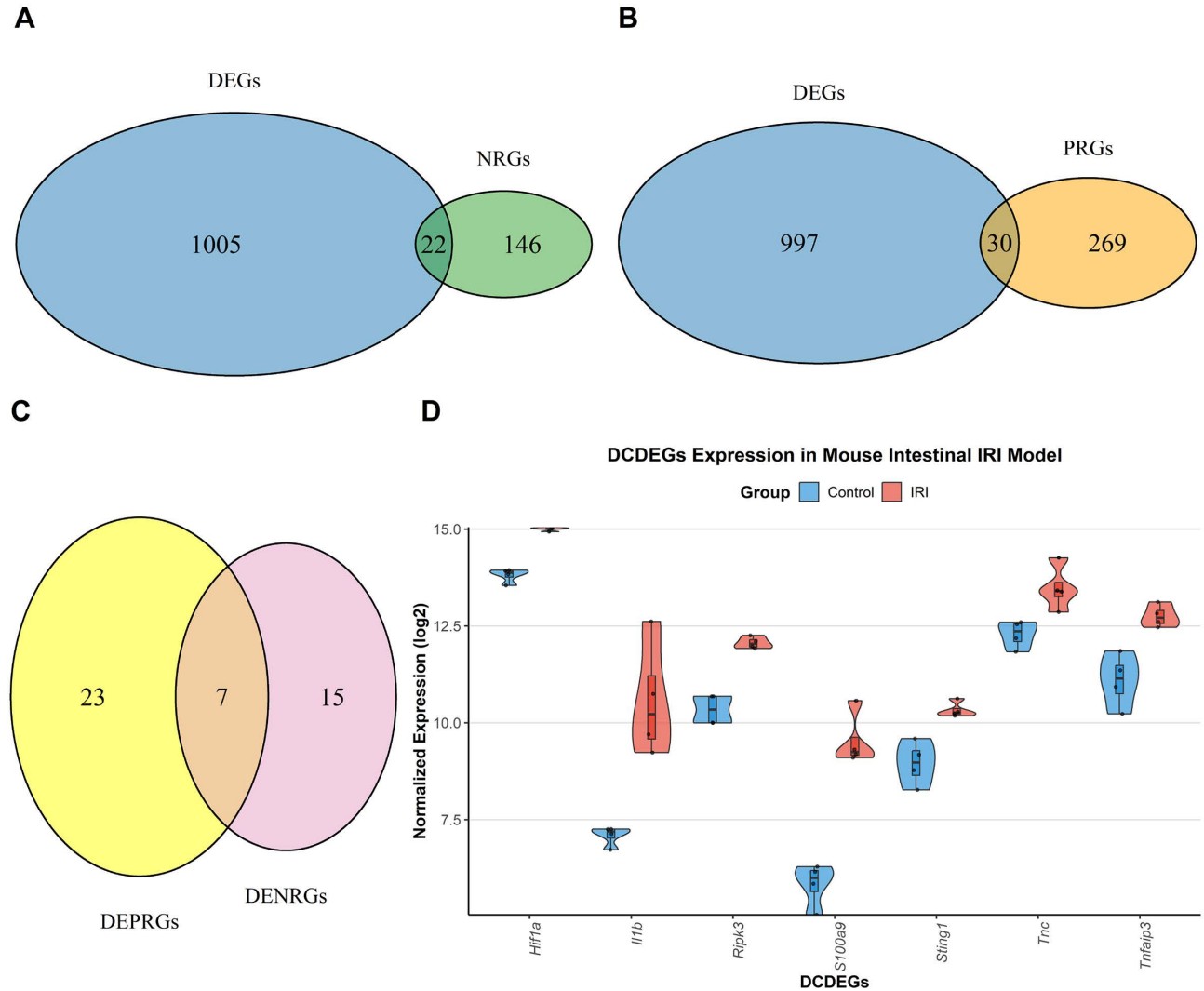

**Fig 3. Identification of DCDEGs in the context of II/R injury. (A)** A Venn diagram illustrates the overlap of gene expression between DEGs and genes related to mouse necroptosis. **(B)** A Venn diagram displays the intersection of genes found in DEGs and those associated with mouse pyroptosis. **(C)** A Venn diagram depicts the overlap of genes between DEPRGs and DENRGs. **(D)** A violin plot represents the expression levels of DCDEGs in Control samples compared to II/R injury samples from GSE96733 (II/R 3 h).

## WGCNA analysis results

To independently validate the hub genes, we performed weighted gene co-expression network analysis (WGCNA) on the GSE96733 (II/R 3 h) training set. After confirming the absence of outlier samples by hierarchical clustering, a soft-thresholding power of 14 was selected to ensure a robust scale-free co-expression network (Fig 6A). Cluster analysis and visualization of sample genes were conducted (Fig 6B). Module-trait relationship analysis then identified the MEturquoise module as the most significantly correlated with the II/R phenotype (Fig 6C). Crucially, all four hub genes were confirmed to be members of this key module, as illustrated by the Venn diagram (Fig 6D), thereby providing independent support for their central role in the II/R response. For detailed information, see S6 Table.

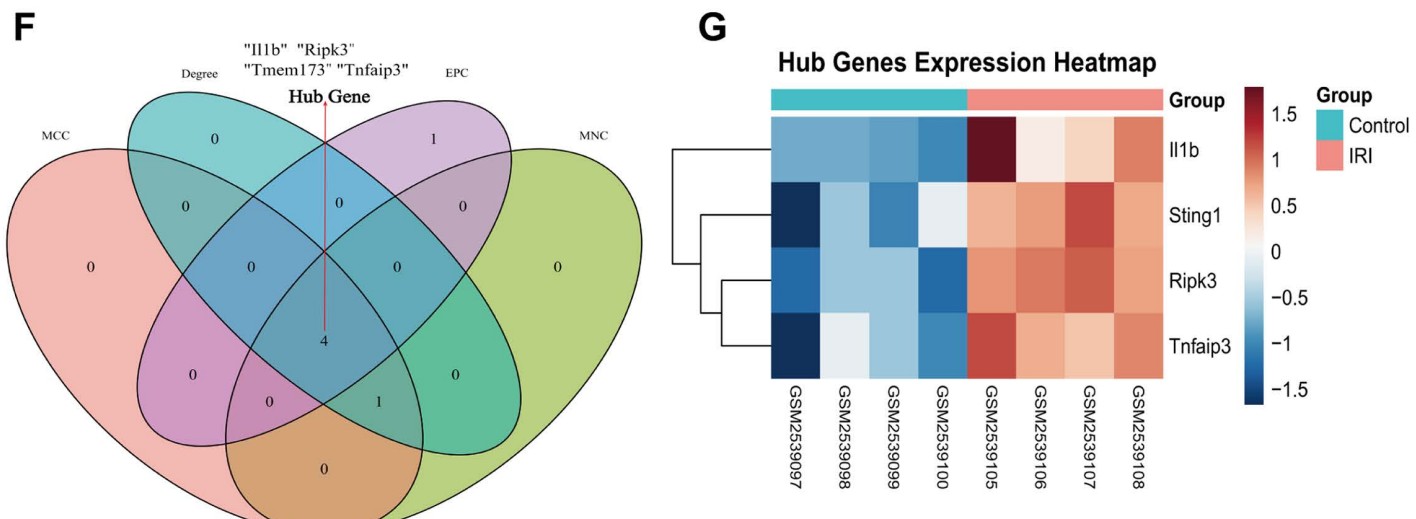

**Fig 4. Identification of hub genes within DCDEGs. (A)** The PPI networks created using STRING. **(B–E)** PPI networks related to DCDEGs were developed through the application of four different algorithms: MCC, MNC, Degree, and EPC. **(F)** By overlapping the top 5 genes identified by each algorithm, four hub genes were determined: *Il1β*, *Ripk3*, *Tmem173 (Sting1)*, and *Tnfaip3*. **(G)** Heatmaps depict the expression levels of the four hub genes in both Control samples and intestinal IRI samples drawn from the GSE96733 (II/R 3 h) dataset.

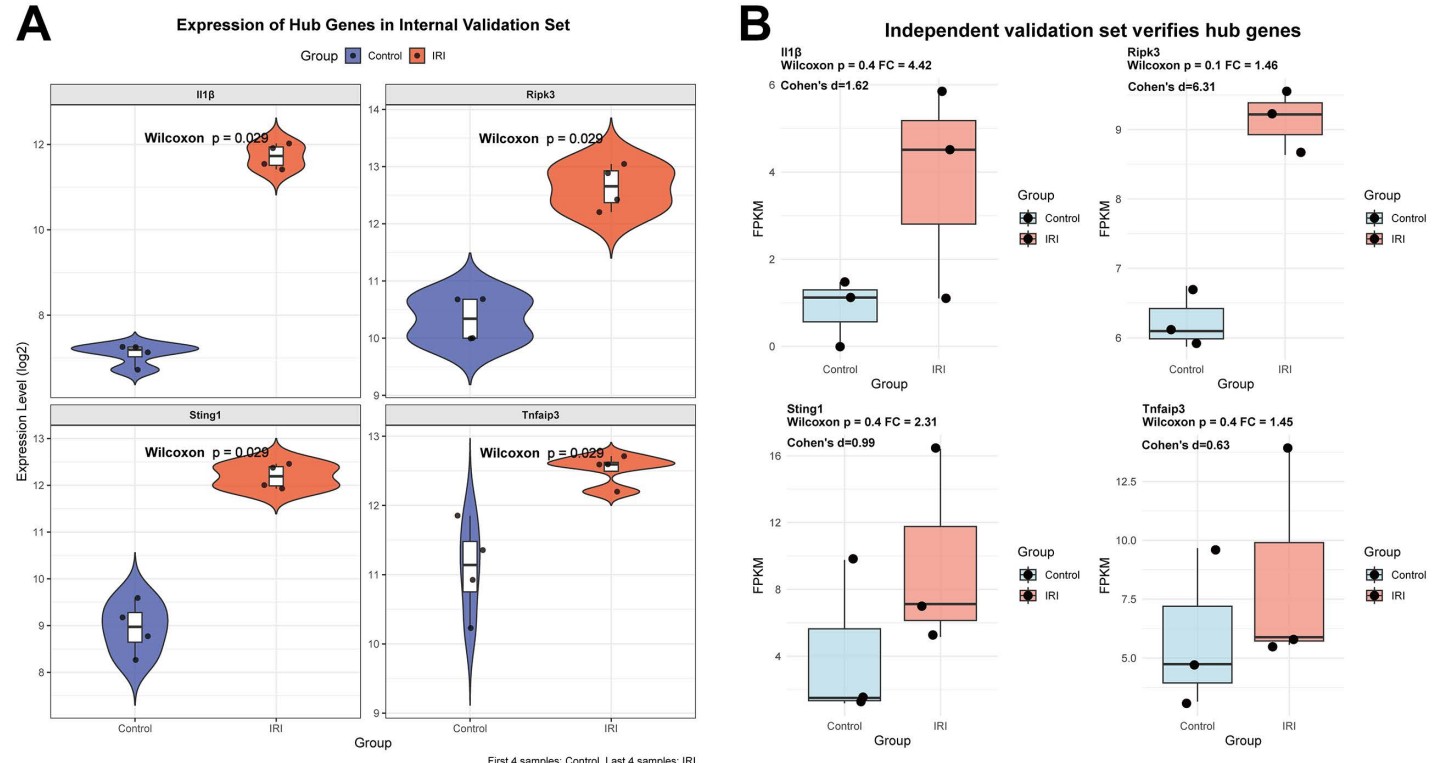

**Fig 5. Verify the transcriptional levels of hub genes in the GSE96733 (II/R 6 h) and GSE232246. (A)** Validate the expression levels of *Il1β*, *Ripk3*, *Tmem173 (Sting1)*, and *Tnfaip3* using the GSE96733 (II/R 6 h) dataset, with results presented as violin plots, $P < 0.05$ indicates statistical significance. **(B)** Independent validation set (GSE232246) verifies hub genes. Box plots are used to visualize validation results. Cohen's d = 0.2: small effect, indicating a difference exists but may lack practical significance; Cohen's d = 0.5: moderate effect, demonstrating substantial significance; Cohen's d = 0.8: large effect, reflecting a marked substantive difference; Cohen's d > 1.0: very large effect, signifying a strong substantive difference.

## Functional enrichment analysis

Gene Ontology (GO) and KEGG pathway enrichment analyses were performed to elucidate the functional roles of the hub genes. The hub genes were significantly enriched in 650 biological processes and 12 KEGG pathways. GO enrichment analysis indicated that the hub genes primarily participated in biological functions such as immune regulation, cell death, and DNA binding, and were located in subcellular structures like autophagosome membranes (Fig 7A). KEGG pathway enrichment analysis evidenced that the central genes were notably overrepresented in pathways including the NOD-like receptor pathway, TNF pathway, and Necroptosis pathway. Notably, hub genes were also enriched in the anti-infection and viral response pathways, suggesting that hub genes may be implicated in intestinal pathogen defense and could be associated with the mechanism of microbiota dysbiosis following intestinal ischemia (Fig 7B). Combining the two enrichment analyses revealed that hub genes are strongly aligned with the pathological mechanisms of necroptosis and pyroptosis, and may mediate their synergistic effects through K63 ubiquitination (GO-MF) and inflammasome (KEGG-NOD pathway). Enrichment in adaptive immunity (GO-BP) and IL-17 pathways (KEGG) indicates the importance of T cell-mediated immune injury in intestinal ischemia-reperfusion. Cell adhesion-related genes (GO-BP) may participate in the repair process following intestinal barrier disruption. For detailed information, see S7 Table.

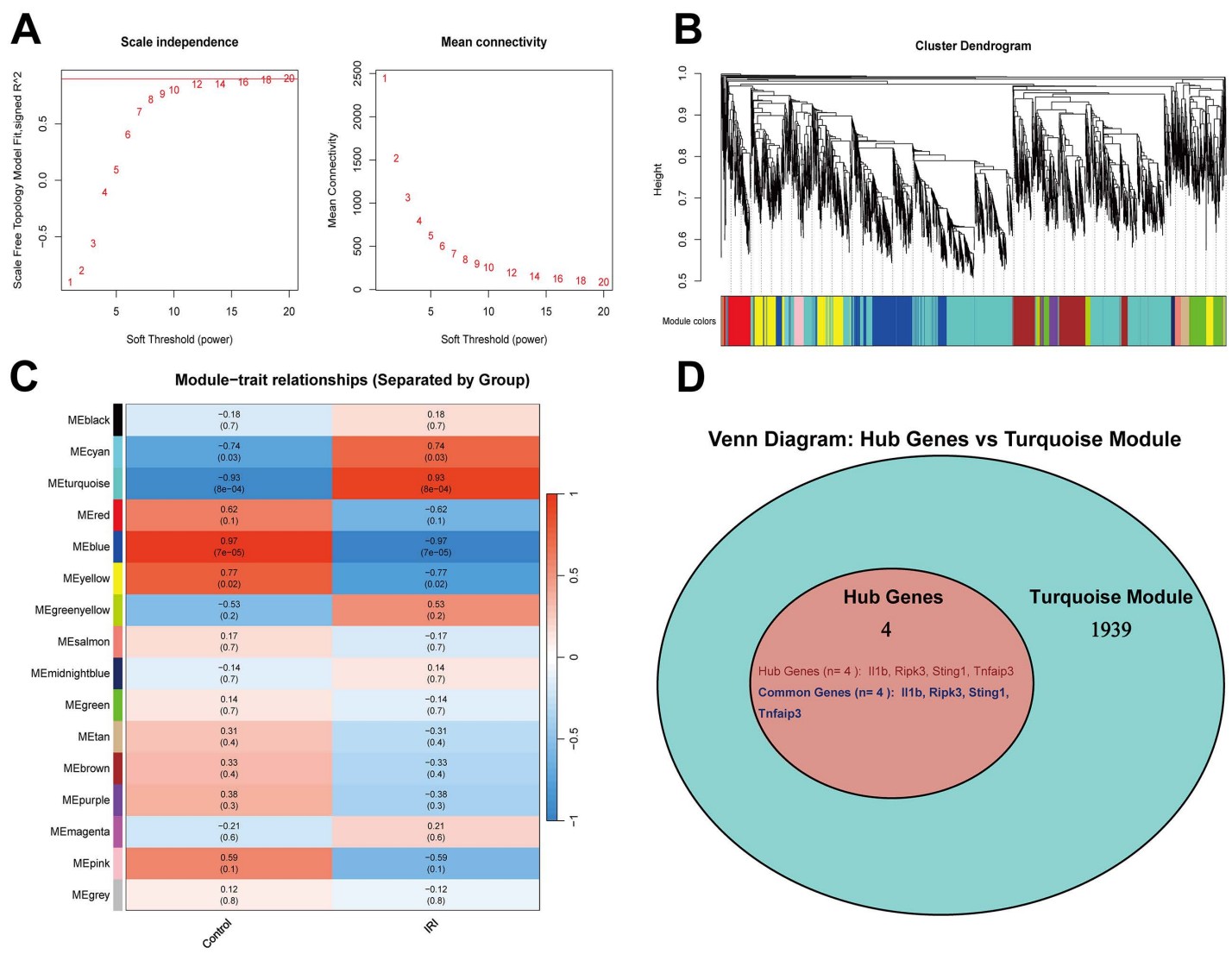

**Fig 6. WGCNA analysis. (A)** The power value scatter plot illustrates appropriate power values. **(B)** Gene clustering and heatmaps reveal genes grouped into distinct modules, strongly reflecting similarities in gene expression patterns. **(C)** Module-trait heatmaps display correlations between different modules and groups. **(D)** Venn diagrams illustrate the intersection between hub genes and key gene modules.

## Immune infiltration analysis

The immune microenvironment in intestinal ischemia-reperfusion (II/R) injury was characterized using CIBERSORT analysis of the GSE96733 training cohort (II/R 3 h). Bar plots illustrate the immune cell composition across samples, while box plots compare immunocyte distributions between control and II/R-damaged groups (Fig 8A and 8B). The II/R injury group exhibited significant alterations in immune cell infiltration patterns, with elevated levels of activated dendritic cells, M2 macrophages, activated NK cells, memory CD4+T cells, naïve CD4+T cells, and Th2 cells compared to controls. Correlation analysis revealed specific hub gene-immune cell interactions underlying this immune remodeling (Fig 8C). *Il1β* and *Ripk3* showed positive associations with mast cells and neutrophils, indicating pro-inflammatory functions. Additionally, *Ripk3* and *Sting1* correlated with eosinophil infiltration, while hub genes collectively demonstrated negative regulation

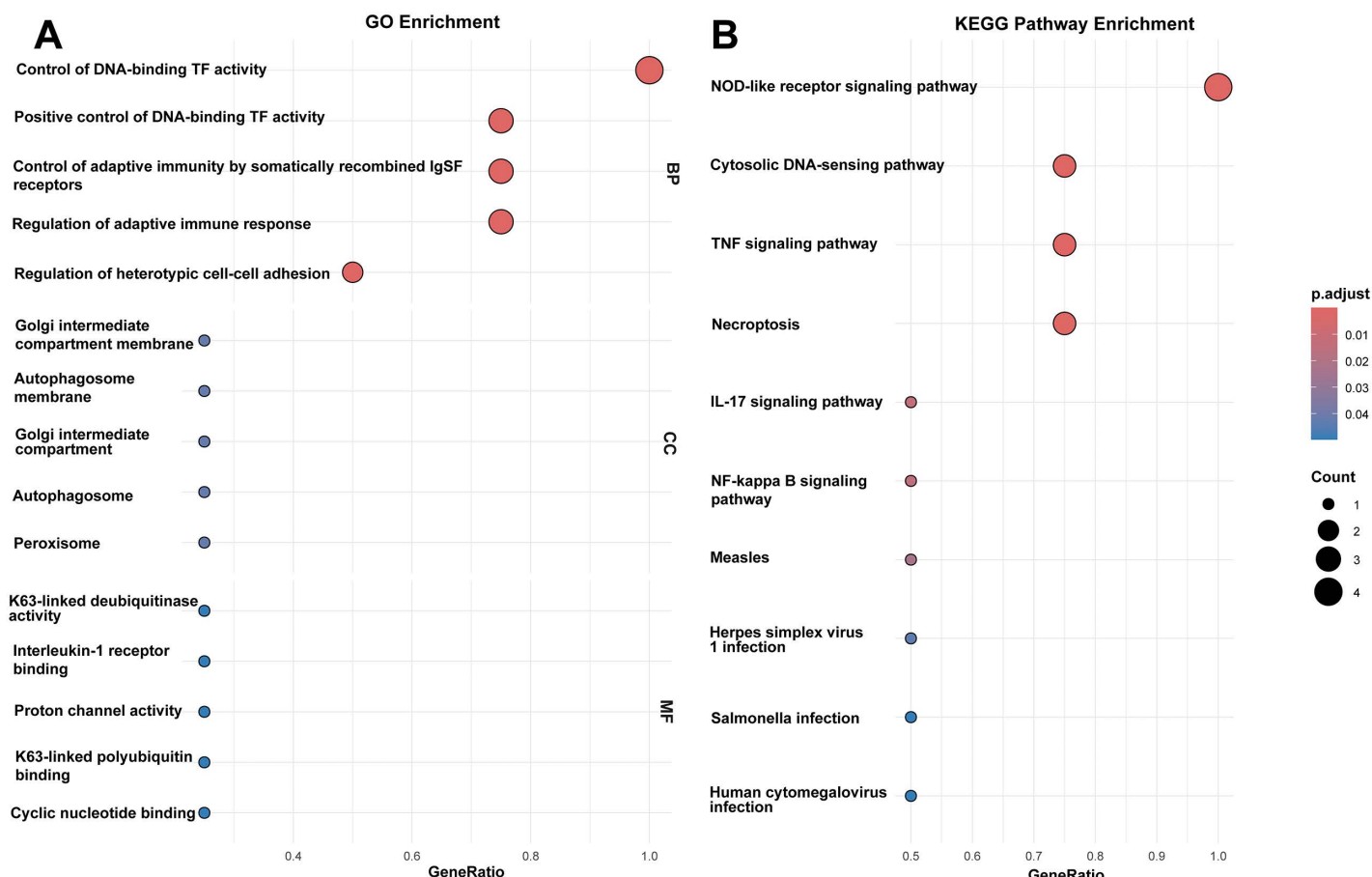

**Fig 7. Analysis of hub gene enrichment.** Enrichment analyses for GO focusing on BP, CC, and MF along with KEGG pathway enrichment, were conducted on the four identified hub genes. **(A)** The bubble chart displays the top 5 enriched entries from the GO analysis. **(B)** The bubble chart displays the top 15 enriched pathways identified through KEGG analysis.

of immature dendritic cells, potentially disrupting immunosuppressive mechanisms and exacerbating inflammatory responses. It is important to note that this immune infiltration analysis is exploratory and was conducted on a small sample dataset. Given these limitations, the results warrant cautious interpretation, particularly since many cell types showed no statistically significant differences. For detailed information, see S8 Table.

## Predictive model building and validation

In this study, we constructed a predictive model for intestinal ischemia-reperfusion injury (IRI) using strong regularized Lasso regression (λ = 0.1) based on limited sample data, identifying key molecular determinants from four candidate hub genes. Feature importance analysis revealed differential contributions among genes: *Ripk3* emerged as the primary predictor (standardized coefficient = 1.763), followed by *Il1β* (coefficient = 0.376) and *Sting1* (coefficient = 0.167), while *Tnfaip3* was Lasso-shrunk to zero, indicating negligible contribution in the current model (Fig 9A). The model demonstrated excellent discrimination in the training set (GSE96733) (AUC = 1.000) and maintained moderate predictive performance in the independent validation set (GSE37013) (AUC = 0.714, 95% CI: 0.450–0.978), suggesting reasonable generalization ability (Fig 9B). Bootstrap resampling analysis (2000 iterations) further confirmed the model's stability, with AUC

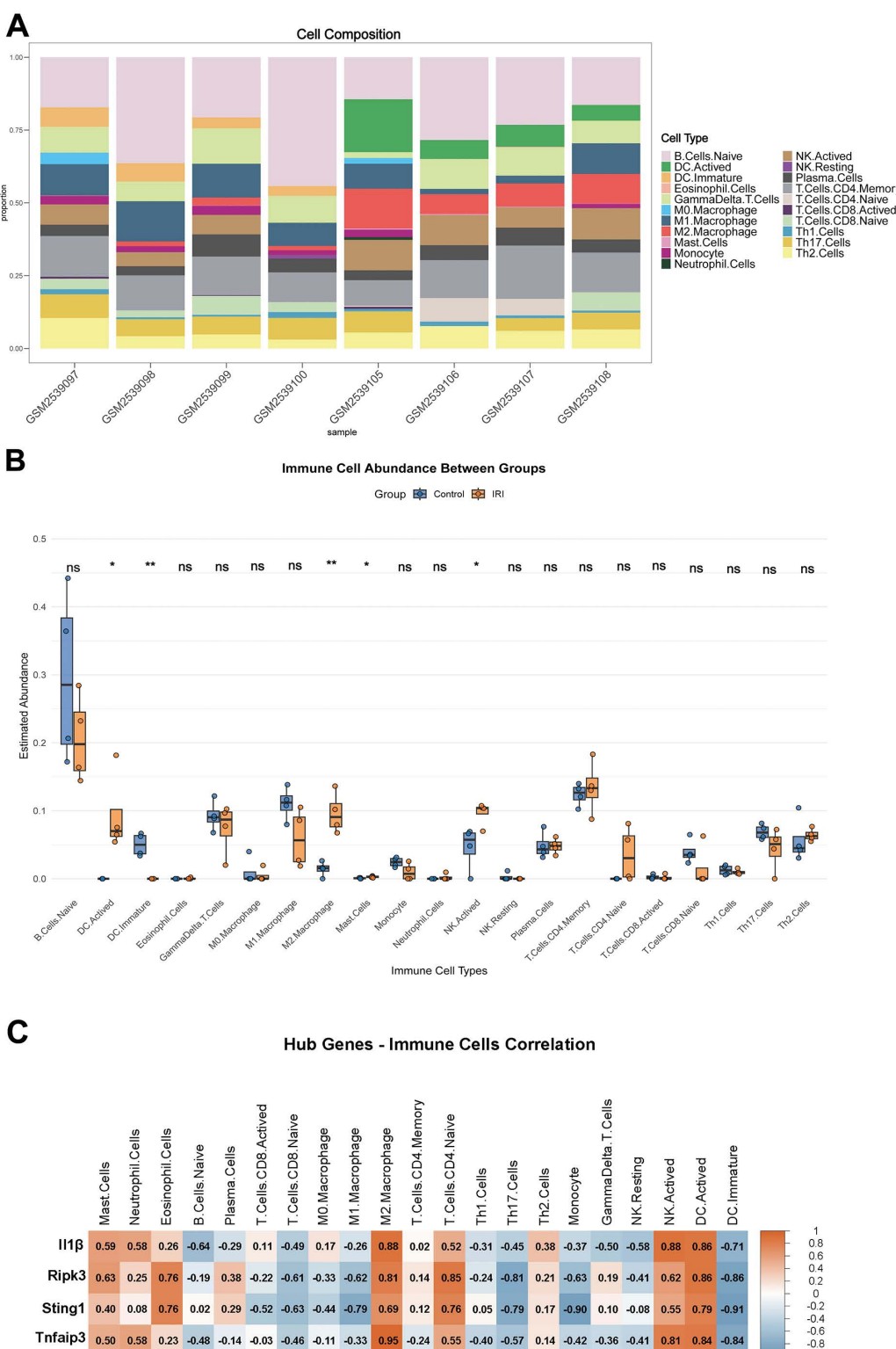

**Fig 8. Analysis of immune infiltration results. (A)** The histograms display the relative proportions of 21 different immune cell types. **(B)** Box plots represent the relative expression levels of each immune cell subtype compared between Control samples and intestinal IRI samples. **(C)** The connection between immune cells and four key genes is illustrated. *$P < 0.05$, **$P < 0.01$, ***$P < 0.001$, ****$P < 0.0001$, ns indicates not significant.

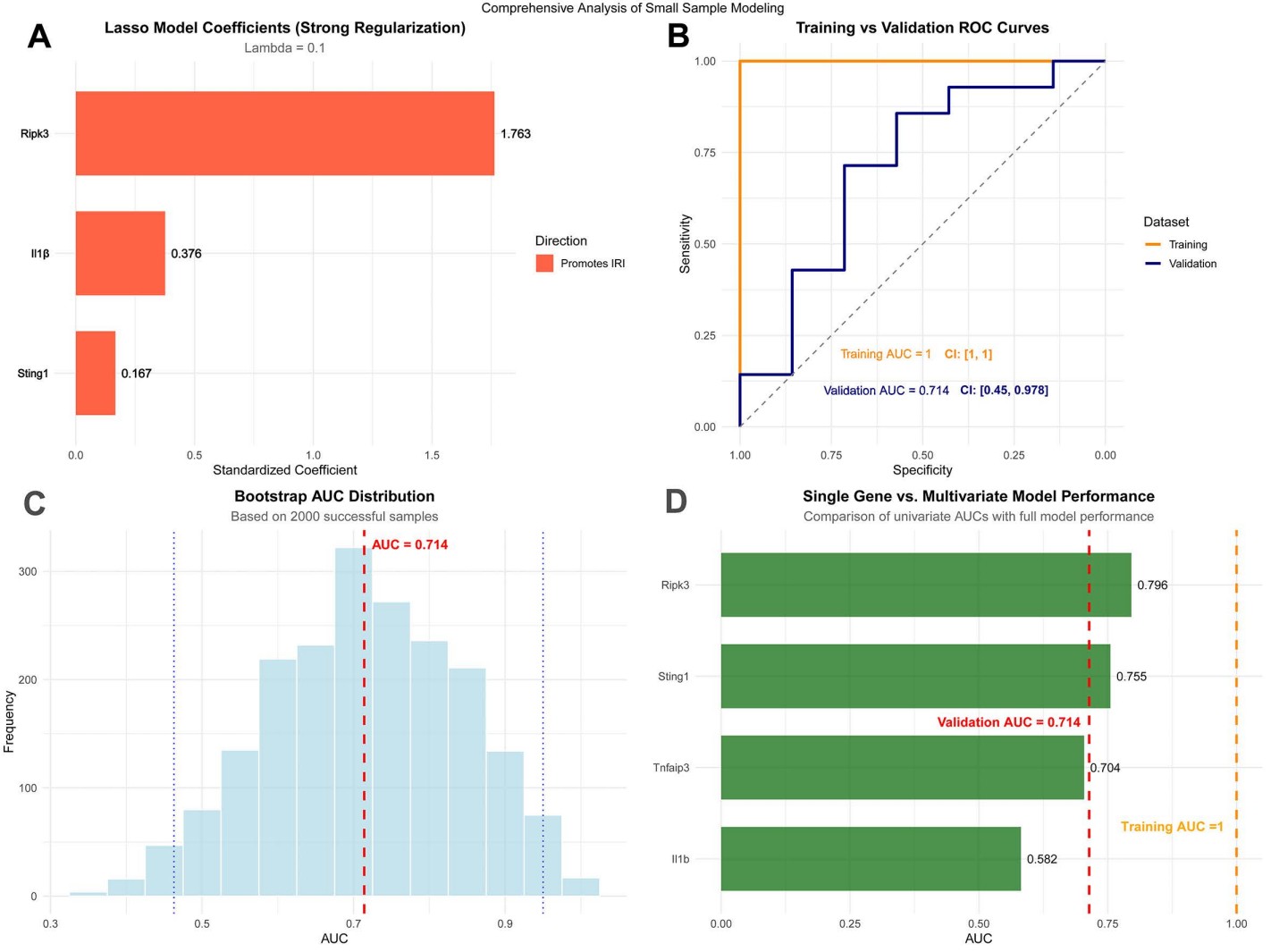

**Fig 9. Development of a predictive model. (A)** Analysis of feature importance for hub genes in predictive models. **(B)** ROC curve for assessing the efficacy of the predictive model. **(C)** Bootstrap resampling analysis evaluates model stability.

values predominantly concentrated within the 0.6–0.8 range (Fig 9C). Comparing univariate and multivariate approaches revealed that despite integrating multi-pathway information with biological plausibility, the multivariate model did not outperform the single-gene *Ripk3* model (AUC = 0.796). This indicates that, under the current small-sample context, introducing more complex multivariate approaches does not enhance predictive efficacy (Fig 9D). Collectively, these findings support *Ripk3* as the most robust single-gene IRI predictor at present. However, given the limited sample size, we believe the core value of this model lies in highlighting the potential for these four pivotal genes to serve as a biomarker combination worthy of further validation in future large-scale studies. For detailed information, see S9 Table.

## Prediction of potential drugs

We used the R package "biomaRt" to map the four hub genes to their human homologs (*IL1β*, *RIPK3*, *STING1*, *TNFAIP3*) and queried them against the DGIdb database, identifying a total of 54 potential therapeutic agents. These agents were

visualized using the "ggplot2" package, presenting an overview of all compounds (Fig 10A) and a gene-drug interaction network (Fig 10B). Furthermore, the top three scoring drugs for each gene were selected and displayed (Fig 10C). The database did not predict any drugs targeting *RIPK3*. For *IL1β*, the predicted agents were gevokizumab, TT-301, and canakinumab; for *STING1*, they were MK-1454 and ADU-S100; and for *TNFAIP3*, the drug methotrexate was identified. For detailed information, see S10 Table.

## Molecular docking validation of drug candidates

To further evaluate the structural feasibility of the predicted drugs, we performed molecular docking analysis on the top-ranked candidates. The 3D docking model revealed that the ligand TT-301 forms two hydrogen bonds with residues Glu80

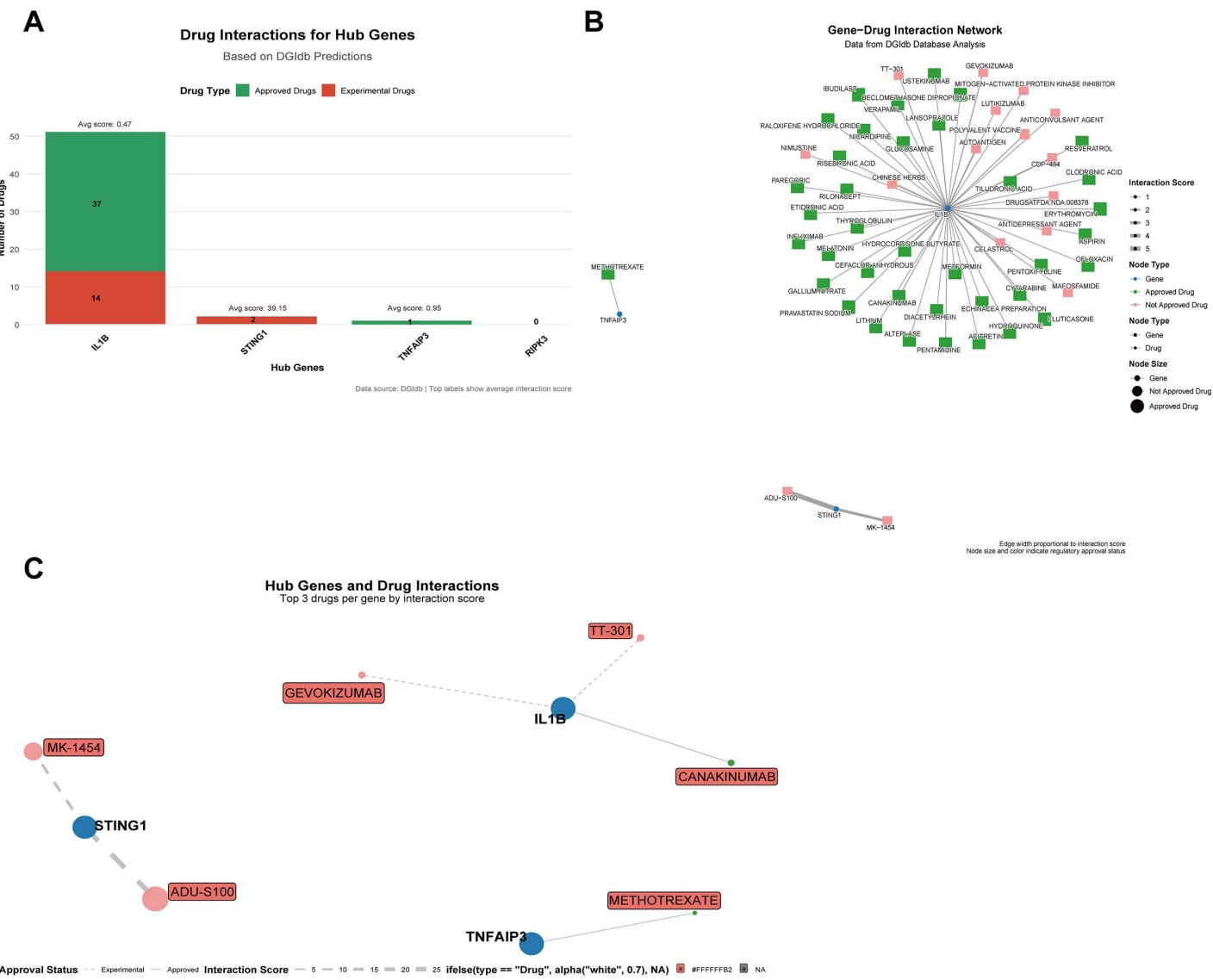

**Fig 10. Drug prediction results. (A)** Bar chart showing 54 predicted drugs. **(B)** Gene-drug network diagram showing all drugs corresponding to each gene. Green indicates approved drugs, while pink indicates unapproved drugs. **(C)** Gene-drug network diagram showing the top 3 drugs for each gene (sorted by score). **(D)** Performance comparison between multivariate models and univariate models.

and Pro78 of the *IL1β* protein, with bond lengths of 2.5 Å and 2.1 Å, respectively, indicating stable binding. The calculated binding energy was −7.7 kcal/mol, suggesting a strong ligand–protein interaction, further stabilized by hydrophobic contacts within the *IL1β* active pocket (Fig 11A). For *STING1*, ligand MK-1454 formed one hydrogen bond (1.9 Å) with Arg238 and multiple additional hydrogen bonds (2.2–2.4 Å) with Thr263, Ser162, Thr267, and Ser241—all located within the protein's activation pocket—implying potential agonist/antagonist functionality (Fig 11B). In comparison, ADU-S100 formed hydrogen bonds with Gly166 and Thr263 of *STING1* (2.6–2.7 Å). Although structurally simpler than MK-1454, it still exhibited robust binding (Fig 11C). The binding energies for MK-1454 and ADU-S100 were −10.8 kcal/mol and −10.2 kcal/mol, respectively, indicating exceptionally high affinity. Between the two, MK-1454 displayed a more elaborate interaction network. Docking of methotrexate with *TNFAIP3* yielded a binding energy of −7.6 kcal/mol, reflecting strong binding. The ligand contacted multiple residues (e.g., Glu305, Tyr306, Leu236, Phe138); however, the absence of short hydrogen bonds (<2.5 Å) suggests possible conformational instability (Fig 11D). In summary, *STING1* stands out as the most promising target in this screening, with MK-1454 exhibiting both strong binding affinity and a well-defined interaction profile. For detailed information, see S11 Table.

### RT-qPCR validation of hub genes

To validate the bioinformatics analysis results, we employed qRT-PCR technology to detect the transcriptional levels of four hub genes in the intestinal ischemia-reperfusion injury model. Compared with the sham-operated group, all four hub genes exhibited significant statistical differences in the II/R injury group. The expression of *Il1β*, *Ripk3*, and *Sting1* was significantly upregulated, while the transcriptional level of *Tnfaip3* was markedly decreased (Fig 12). Among these, the

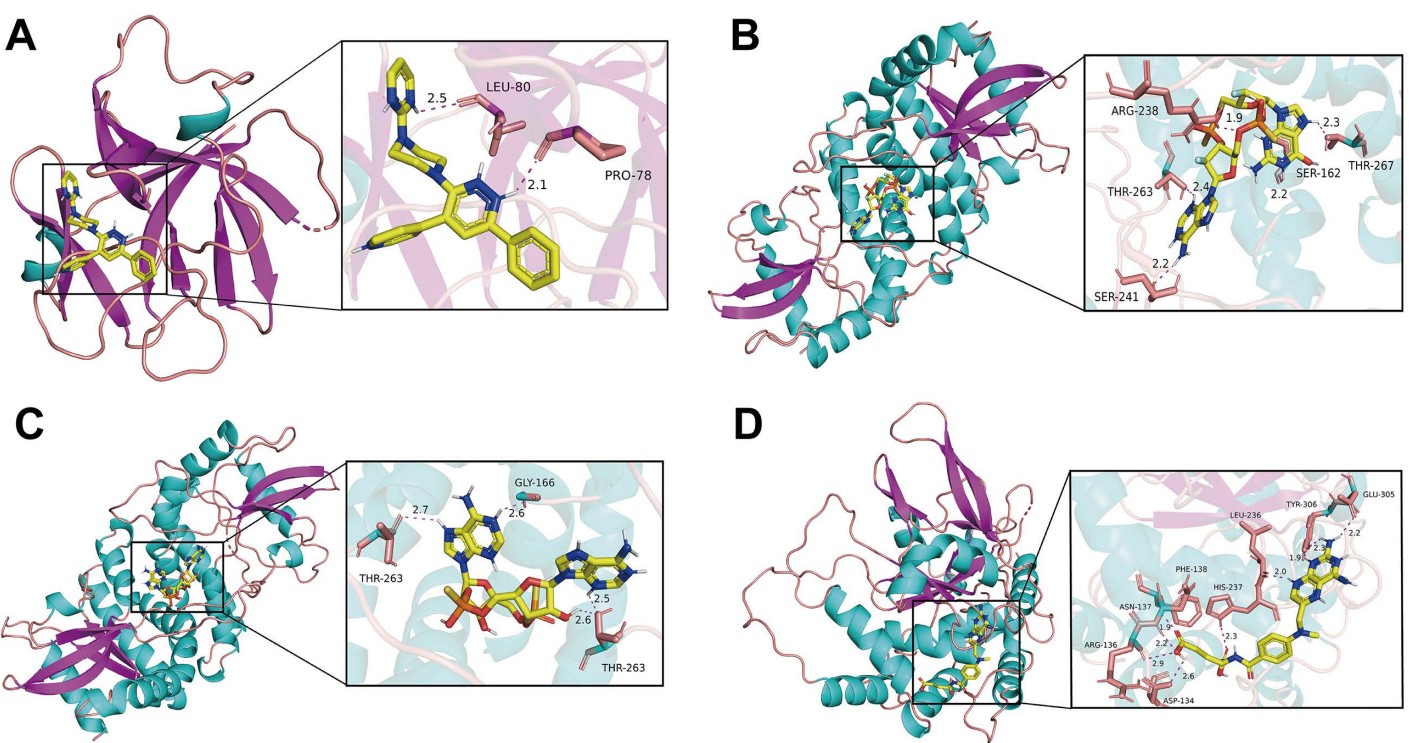

**Fig 11. Molecular docking results. (A)** Docking results of *IL1β* with TT-301. **(B)** Docking results of *STING1* with MK-1454. **(C)** Docking results of *STING1* with ADU-S100. **(D)** Docking results of *TNFAIP3* with methotrexate.

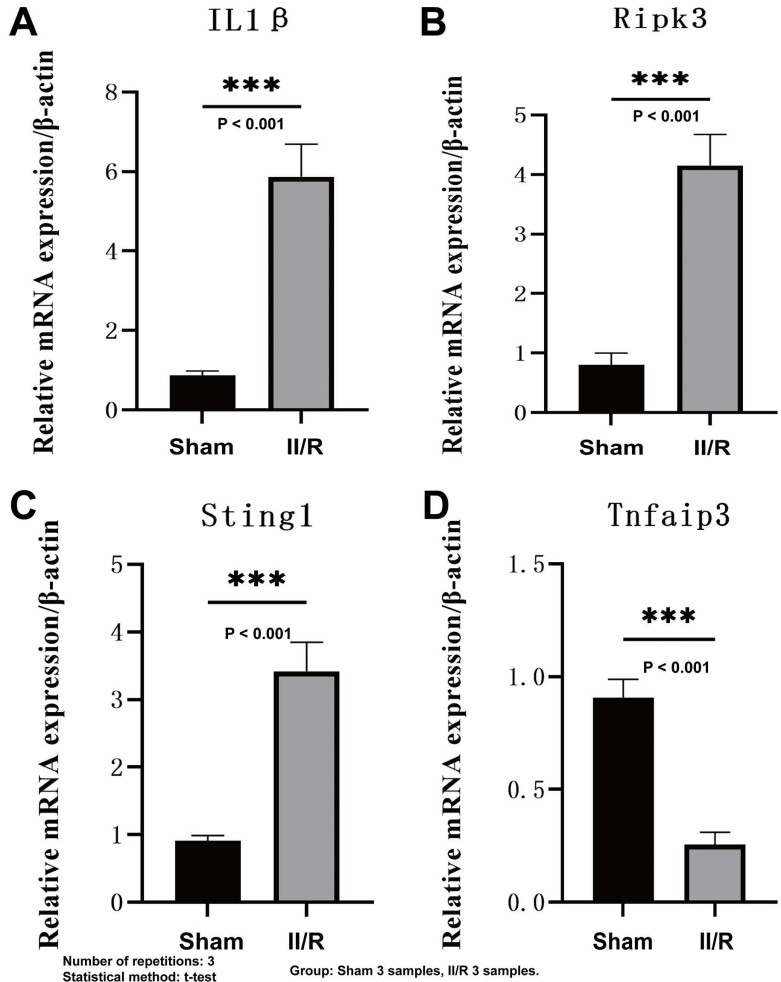

**Fig 12. Asessment of central hub genes expression in a mouse model of II/R injury.** Sample size [6], three experimental replicates, and statistical test method (t-test) **(A–D)** The mRNA expression levels of *Il1β*, *Ripk3*, *Sting1*, and *Tnfaip3* were analyzed through qRT-PCR. *P < 0.05, **P < 0.01, ***P < 0.001, ****P < 0.0001, ns indicates not significant.

expression trends of the first three genes were consistent with bioinformatics analysis results, but the expression pattern of *Tnfaip3* differed from bioinformatics evaluation. For detailed information, see S12 Table.

## Discussion

This study systematically investigated the expression patterns and functional associations of genes related to necroptosis and pyroptosis in intestinal ischemia-reperfusion (II/R) injury, aiming to elucidate their potential roles in the pathogenesis of II/R. By integrating multiple bioinformatics approaches, we identified four hub genes—*Il1β*, *Ripk3*, *Sting1*, and *Tnfaip3*—that may constitute a key molecular hub regulating cell death and inflammatory responses. Consistent upregulation of *Il1β*, *Ripk3*, and *Sting1* was observed across the training dataset, WGCNA modules, and in vivo qPCR validation, suggesting their potential as biomarkers for II/R injury. Although *Tnfaip3* exhibited inconsistencies between bioinformatics predictions and experimental validation, this discrepancy may reflect its dynamic regulation during injury or cell-type-specific functions warranting further investigation [25,26].

From a mechanistic perspective, *IL1β* is a pro-inflammatory cytokine that, in the intestine, directly causes intestinal barrier disruption by activating neutrophils and promoting oxidative stress, and further triggers damage to distal organs [27,28]. *RIPK3* is a key executor of necrotic apoptosis, triggering cell membrane rupture by phosphorylating *MLKL* [29], while also participating in inflammasome activation, *IL1β* secretion, and vascular stability regulation [30]. *RIPK3* deficiency may alleviate inflammation by inhibiting *NF-κB* and *STAT3* signaling, but its direct role in II/R injury has been poorly studied [31]. *STING1* is a cytoplasmic DNA sensor that recognizes DNA released by pathogens or damage (such as mtDNA), activating the type I interferon (*IFN*) pathway and the *NLRP3* inflammasome [32]. mtDNA release activates *STING1*, which promotes macrophage pyroptosis (*Caspase-1*/*GSDMD*-dependent) via the cGAS-cGAMP pathway. *STING1* induces calcium ion influx, exacerbating endoplasmic reticulum stress and mitochondrial damage [33]. *TNFAIP3* (tumor necrosis factor α-induced protein 3, A20) negatively regulates the *NF-κB* signaling pathway, inhibiting inflammation and autoimmune responses [25]. In the I/R model, *TNFAIP3* deficiency leads to barrier disruption and microbiota translocation [34]. Therefore, we believe that *Il1β*, *Ripk3*, and *Sting1* may synergistically form a signaling axis that collectively drives inflammatory amplification and cell death processes during I/R injury [35]. We hypothesize that during ischemia, mitochondrial DNA release into the cytoplasm may activate *Sting1* signaling [32,33], thereby promoting *Ripk3*-mediated necroptosis and *Il1β*-driven pyroptosis [27,29]. This creates a positive feedback loop that exacerbates tissue injury and the inflammatory cytokine storm. To explain the observed paradox in *Tnfaip3* expression, we propose a hypothesis: the pro-inflammatory signaling via the *STING1-RIPK3-IL1β* axis may simultaneously induce *Tnfaip3* transcription while also triggering rapid ubiquitin-mediated degradation of its protein product, possibly through inflammation-associated E3 ligases [36,37]. This would lead to elevated mRNA levels but insufficient functional protein accumulation, resulting in the apparent negative correlation between its transcriptional readout and phenotypic outcome. Conversely, *Tnfaip3* may exert negative regulation by inhibiting pro-inflammatory pathways like *NF-κB* through its ubiquitin-editing capacity, thereby promoting inflammation resolution and tissue repair in the late phase of injury [25,34]. This dual regulatory network of positive and negative control likely jointly determines the intensity and outcome of the inflammatory response during II/R(25).

Immune cell infiltration analysis further supports the aforementioned inference. The increased proportion of M2 macrophages, dendritic cells, activated NK cells, and CD4⁺T cells in II/R samples suggests that both innate and adaptive immune responses are involved in the injury process. Correlation analysis revealed that *Il1β* and *Ripk3* positively correlated with M2 macrophages and mast cells, implying their potential synergistic role in maintaining the inflammatory microenvironment; *Sting1* showed positive correlations with CD4⁺naïve T cells and negative correlations with monocytes, suggesting its pivotal role in immune cell crosstalk. The significant correlation between *Tnfaip3* and M2 macrophages aligns with its anti-inflammatory and tissue-repair-promoting functions. Collectively, these findings reveal the intricate interplay between cell death pathways and immune regulatory networks in II/R injury.

In terms of translational value, drug-gene interactions and molecular docking analysis have identified potential intervention targets. Several drugs targeting these pivotal genes have entered the research landscape, such as canakinumab and gevokizumab (targeting *IL1β*) [38,39], MK-1454 and ADU-S100 (targeting *STING1*) [40], and methotrexate (associated with *TNFAIP3* regulation) [41]. Most of these drugs possess distinct immunomodulatory activities and may exert therapeutic effects by intervening in the "necroptosis-pyroptosis-inflammation" network. For instance, inhibiting *STING1* may simultaneously alleviate *RIPK3*- and *IL1β*-mediated injury responses, while enhancing *TNFAIP3* function could aid in restoring immune homeostasis. Although these findings remain largely predictive at present, they offer novel insights for developing precision intervention strategies targeting II/R.

This study still has several limitations. The bioinformatics analysis, which focused on intersecting genes associated with necroptosis and pyroptosis, may have overlooked certain important "non-overlapping" genes such as *Il1a*, potentially limiting the comprehensiveness of our findings. Moreover, the sample size relied upon for bioinformatics and experimental validation was relatively small, and extrapolating findings from mouse models to human pathology requires caution.

Whole-organism RNA analysis may fail to reveal cell type-specific expression profiles, which could be one reason for the apparent contradictions in gene expression observed for genes such as *Tnfaip3*. Future studies should validate the expression and function of these genes in larger cohorts and human tissue samples. Utilizing gene editing and cell-specific knockout approaches will further elucidate the molecular pathways of the *Sting1–Ripk3–Il1β–Tnfaip3* regulatory axis in II/R. Additionally, conducting in vivo pharmacological experiments targeting these pathways will be a critical step in assessing their clinical translational value.

## Conclusion

Our study nominates *Il1β*, *Ripk3*, *Sting1*, and *Tnfaip3* as plausible mediators of cell-death–inflammation cross-talk in II/R injury and provides a curated short list of drug–gene interactions to prioritize in follow-up experiments. Given sample-size constraints, we position these results as a framework for targeted validation rather than definitive biomarkers.

## Supporting information

**S1 Table. R code.**
(DOCX)

**S2 Table. Analysis results of DEGs.**
(XLS)

**S3 Table. Identification of genes related to cell death.**
(XLS)

**S4 Table. Hub gene screening results.**
(XLS)

**S5 Table. Hub gene validation.**
(XLS)

**S6 Table. WGCNA Analysis.**
(XLS)

**S7 Table. Functional enrichment analysis results.**
(XLS)

**S8 Table. Immune infiltration analysis results.**
(XLS)

**S9 Table. Predictive model building and validation.**
(XLS)

**S10 Table. Drug prediction results.**
(XLS)

**S11 Table. Molecular docking results.**
(XLS)

**S12 Table. Results of RT-qPCR validation of key genes.**
(XLS)

## Acknowledgments

The authors appreciate the editor and all reviewers for their time spent in adding professional remarks and making constructive suggestions.

## Author contributions

**Data curation:** Shuang Bao, YanBo Sun, XueFen Lei, YuanPei Zhao.

**Formal analysis:** Shuang Bao.

**Funding acquisition:** Weiming Li.

**Resources:** Weiming Li.

**Visualization:** YanBo Sun, YiChen Hu, XueFen Lei.

**Writing – original draft:** Shuang Bao, YiChen Hu, XueFen Lei.

**Writing – review & editing:** Shuang Bao, YanBo Sun, YiChen Hu, YuanPei Zhao, Weiming Li.

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
