## [Decision Letter · Decision Letter 0]

16 Oct 2025

Dear Dr. Li,

Thank you for submitting your manuscript to PLOS ONE. After careful consideration, we feel that it has merit but does not fully meet PLOS ONE’s publication criteria as it currently stands. Therefore, we invite you to submit a revised version of the manuscript that addresses the points raised during the review process.

We look forward to receiving your revised manuscript.

Kind regards,

Tomasz W. Kaminski

Academic Editor

PLOS ONE

Journal Requirements:

3. Please note that PLOS One has specific guidelines on code sharing for submissions in which author-generated code underpins the findings in the manuscript. In these cases, we expect all author-generated code to be made available without restrictions upon publication of the work. Please review our guidelines at https://journals.plos.org/plosone/s/materials-and-software-sharing#loc-sharing-code and ensure that your code is shared in a way that follows best practice and facilitates reproducibility and reuse.

 “The author(s) declare that financial support was received for the research and/or publication of this article. This work was funded by the National Natural Science Foundation of China (NSFC) (No. 82460114) and foreign cooperative research project of the Second Affiliated Hospital of Kunming Medical University (No.2022dwhz09).”

5. Please note that funding information should not appear in any section or other areas of your manuscript. We will only publish funding information present in the Funding Statement section of the online submission form. Please remove any funding-related text from the manuscript.

Reviewer's Responses to Questions

**Comments to the Author**

1. Is the manuscript technically sound, and do the data support the conclusions?

Reviewer #1: Partly

Reviewer #2: Partly

2. Has the statistical analysis been performed appropriately and rigorously?

Reviewer #1: Yes

Reviewer #2: No

3. Have the authors made all data underlying the findings in their manuscript fully available?

Reviewer #1: Yes

Reviewer #2: Yes

4. Is the manuscript presented in an intelligible fashion and written in standard English?

Reviewer #1: Yes

Reviewer #2: Yes

Reviewer #1: Bao, Sun, Hu, Lei, et al. share here their thorough exploration of differentially expressed, cell death related transcripts in both human and murine intestinal ischemia reperfusion injury. While the authors’ findings are far from surprising, they are critical data in addressing the urgent need for new and more selective targets in addressing systemic inflammation such as occurs in I/IRI. The manuscript is well-written and presented for the most part, and will hopefully prove a strong contribution to the field, however I have a number of both major and minor critiques that must be addressed prior to publication.

Major Comments:

- There are obviously quite a few different mechanisms of cell death, all of which are known to occur in ischemia reperfusion injury. Please better justify limiting your analysis to necroptosis and pyroptosis while neglecting, say, apoptotic or ferroptotic genes.

- It is unclear to me why you are looking only at the genes in both the necroptosis- and pyroptosis-related gene categories? Il1a doesn’t stop being a differentially expressed, cell death related gene just because it’s on the list of Necroptotic genes but not Pyroptotic genes. Please clarify your decision to limit the pool of genes like this, or else include analysis of all DENRGs and DEPRGs.

- The predictive model presented in figure 7 is of very low accuracy and reliability and its inclusion in the manuscript detracts from your findings more than it adds anything. I’d suggest removing it.

- It is unclear from either the methods or results sections where and how the samples examined in Figure 10 through qRT-PCR were acquired and isolated. This is critical information, please include it.

- Relatedly, given that you are able to perform gene expression analysis, I would very much like to see immunoblot analysis as well. RIPK3 may be upregulated, but whether it’s being cleaved or not is a critical question in regards to its activity, which can be visualized by western blot.

- In the end of your discussion, you state “this study has obvious limitations.” Please actually state what these limitations are.

Minor Comments

- In the Introduction, you often introduce a concept like so: “Reperfusion phase: ROS trigger…” or “Intestinal barrier disruption: Intestinal villus tip…” This formatting is confusing, and I would suggest maintaining standard sentence construction.

- Several points in the methods (e.g. line 100, lines 139-141, etc) are written as instructions rather than a report of what as performed. Please make sure to edit these, for clarity.

- Are the 4 control samples in the validation set the same as the 4 controls in the training set? I would hope not, but if so you definitely need new data for the validation set.

- Please clarify how you collected necroptosis and pyroptosis related genes from the NCBI database.

- Please include vendor as well as product or catalog number in your methods.

- Figure 6C is quite information-dense and visually difficult to parse. Please revise.

- I would like at least a little bit of mechanistic hypothesizing as to why you saw the inverse relationship with Tnfaip3 in your qRT-PCR results.

- Line 58-59: this phrasing “In recent years…” makes it sound like the role of cell death in I/IRI is a new phenomenon in nature. I’d suggest editing to say “In recent years, the role of […] has become apparent.” or similar.

- There’s no need to write drug names in all-capital letters.

- There’s no need to capitalize “Mast Cells”

- Line 289: please replace “DC Immature” with “immature dendritic cells”

Reviewer #2: This manuscript provides a comprehensive bioinformatics and limited experimental study of necroptosis and pyroptosis pathways in intestinal ischemia-reperfusion (II/R) injury. However, there are some gaps in the manuscripts which should be addressed before final publication.

1.Major Comments:

Experimental Design and qRT-PCR Methodology:

The manuscript does not describe how the intestinal I/R injury was induced in mice prior to RNA extraction (see Results lines 258–263, Methods section before line 170).

The procedure for ischemia induction (artery clamped, duration, reperfusion period, anesthesia, euthanasia) must be clearly detailed to ensure clarity as well as reproducibility.

The lack of this information prevents readers from understanding the biological context of the “I/R samples” used for qPCR.

In Methods lines 170–177, the authors state that RNA was extracted and cDNA synthesized, but they never clarify from which tissue or cell type (whole intestinal tissue, scraped mucosa, or isolated epithelial/immune cells). This is a major gap because:

Whole-tissue RNA may mask or invert cell-specific expression profiles (e.g., the Tnfaip3 discrepancy in lines 260–263, 315–320).

Reproducibility and interpretation of gene expression changes depend on the source of RNA.

The authors must specify tissue origin, sampling site (segment of intestine, distance from clamp), time after reperfusion, preservation method, and number of biological and technical replicates.

If only whole-tissue homogenates were used, this limitation should be acknowledged explicitly in Discussion lines 344–349.

Without these methodological details, the qRT-PCR validation cannot be considered reliable.

Validation of Hub Genes:

qRT-PCR validation (lines 257–263) includes only four genes; Tnfaip3 shows an opposite trend. Provide reasoning (cell specificity, time-dependent effects).

Add at least partial protein-level validation (Western blot/IHC) for two hub genes (RIPK3, STING1) or validate with an independent dataset (GSE62198, GSE86618).

Include sample size (n), replicates, and statistical tests in figure captions (Fig. 10)

Predictive Model Reliability:

Predictive model (lines 158–164, 240–247) shows AUC = 1.0 in training and 0.673 in external validation—clear overfitting.

Apply k-fold cross-validation or bootstrapping and report confidence intervals.

Disclose model coefficients and normalization method.

Sample Size and Statistical Power:

Dataset size (lines 102–106, 240–247) is limited (n = 8 training; n = 21 validation).

Emphasize this limitation in Discussion (lines 344–349).

Consider permutation testing to improve robustness.

Immune infiltration Analysis:

CIBERSORT analysis (lines 151–157, results 226–239) on small sample size lacks reliability.

Report deconvolution p-values and apply FDR correction.

Clarify legends in Fig. 7A–C (cell type colors, significance notation).

Interpretation and Causal Inference:

Mechanistic statements (e.g., “STING1–RIPK3 axis constitutes an inflammatory loop”, lines 327–341) are speculative.

Rephrase as hypotheses: “may constitute” or “suggests the involvement of.”

Separate correlative findings from mechanistic inference.

Drug Prediction Analysis:

DGIdb-based predictions (lines 165–169, 248–256) are computational only.

Clarify that they are predicted interactions; provide supporting literature for drugs like Canakinumab or Methotrexate if relevant.

2. Minor Comments

Language and Grammar:

o Introduction (lines 37–97) is verbose; condense overlapping sentences.

o Replace “were overlapped” → “were intersected” (lines 18–21, 121–123).

o Ensure consistent gene nomenclature (Il1b → IL1β; lines 28, 201, 259).

Figures and Legends:

o Add statistical notations (e.g., p < 0.05) to Figs 2–10 (legends lines 475–515).

o Define all abbreviations at first mention (DENRGs, DEPRGs, DCDEGs; lines 18–23, 120–139).

Methods Transparency:

o Clarify normalization for GEO datasets (lines 100–113).

o Ensure GEO accession numbers and supplementary R script (Table S10) are available.

o Add missing details of I/R model and qPCR sampling as outlined in 1.1.

Discussion Organization:

o Current Discussion (lines 264–349) can be divided into:

Bioinformatics summary, Biological interpretation, and Limitations/future perspectives

Ethical and Funding Statements:

o Ethics (lines 359–362)—confirm inclusion of anesthesia, analgesia, and euthanasia details in Methods (around line ~99).

**Do you want your identity to be public for this peer review?** For information about this choice, including consent withdrawal, please see our Privacy Policy

Reviewer #1: **Yes: ** Tzvi Pollock

Reviewer #2: No

---

## [Author Response · Author response to Decision Letter 1]

7 Nov 2025

Response to Reviewers

Reviewer #1

Major Comments:

1. There are obviously quite a few different mechanisms of cell death, all of which are known to occur in ischemia reperfusion injury. Please better justify limiting your analysis to necroptosis and pyroptosis while neglecting, say, apoptotic or ferroptotic genes.

We appreciate the reviewers' comments. Necroptosis and pyroptosis share a common upstream signaling pathway distinct from other forms of cell death, and their terminal effects are directly linked to the core feature of ischemia-reperfusion injury—a severe inflammatory response. Both necroptosis and pyroptosis can be triggered by the activation of pattern recognition receptors. Both represent lytic forms of cell death. They lead to rupture of the cell membrane, releasing large amounts of intracellular contents, including DAMPs, cytokines, and others, thereby forming a potent “positive feedback loop” that dramatically amplifies local and systemic inflammatory responses. This closely parallels the tissue destruction and dysfunction observed after ischemia-reperfusion. Therefore, from a therapeutic perspective, inhibiting these two forms of cell death may more effectively “break” the vicious cycle of inflammation.

2. It is unclear to me why you are looking only at the genes in both the necroptosis- and pyroptosis-related gene categories? Il1a doesn’t stop being a differentially expressed, cell death related gene just because it’s on the list of Necroptotic genes but not Pyroptotic genes. Please clarify your decision to limit the pool of genes like this, or else include analysis of all DENRGs and DEPRGs.

We appreciate the reviewers' comments. Our decision to focus on genes present in both the necroptosis and pyroptosis gene sets (i.e., the intersection) is based on a clear scientific hypothesis: these genes may represent “core regulatory nodes” coordinating these two forms of lytic cell death, potentially playing a particularly crucial role in inflammatory cell death during ischemia-reperfusion injury. Although necrosis and pyroptosis are distinct pathways, they may share upstream signaling or exhibit extensive crosstalk within the specific context of ischemia-reperfusion injury. By screening “intersection genes,” we aimed to prioritize molecules most likely to mediate this crosstalk from the vast array of differentially expressed genes. This approach helps reveal the core of regulatory networks rather than being overwhelmed by numerous collateral effect genes. Analyzing “intersections” constitutes a rigorous screening strategy that significantly enhances the specificity of findings. A gene included in two independent, authoritative cell death gene sets substantially increases its likelihood of being a key molecule in cell death (particularly lytic death). This allows us to focus on the most reliable targets, reducing the risk of false positives in subsequent validation experiments. Furthermore, cell death pathways exhibit substantial redundancy in ischemia-reperfusion injury. Analyzing differentially expressed genes in a single pathway (e.g., necrotic apoptosis) may identify numerous effector molecules whose functions could be compensated by other pathways. Genes capable of simultaneously influencing two critical pathways may yield more potent, less compensable therapeutic effects when targeted. Finally, we fully acknowledge the importance of Il1a. However, in the current study, we prioritized reporting genes within the “intersection” that may possess broader regulatory functions. We explicitly mention important “non-intersection” genes like Il1a in the Discussion section as key directions for future research and complementary to our current model. In summary, we do not dismiss the value of other DENRGs or DEPRGs. Instead, we adopted a “core-to-periphery” research approach. By first deeply analyzing the core gene set connecting the two major pathways, we aimed to yield more insightful discoveries. We believe this strategy provided our study with clear focus and depth.

3.The predictive model presented in figure 7 is of very low accuracy and reliability and its inclusion in the manuscript detracts from your findings more than it adds anything. I’d suggest removing it.

We appreciate the valuable comments provided by the reviewers. To enhance the predictive performance and robustness of the model, we constructed a classification model using strong regularized Lasso regression (λ = 0.1) based on the expression data of four hub genes in the mouse training set, effectively controlling the risk of overfitting in small-sample scenarios. Although the model demonstrated excellent performance on the training set (AUC = 1.0), suggesting potential overfitting, it still achieved moderate predictive performance (AUC = 0.714) on the external validation set, representing a significant improvement over the unmodified model. After careful evaluation, we concluded that the model retains certain reference value at this stage and decided to retain it. Furthermore, we explicitly stated in the paper that this model should be regarded as a preliminary, exploratory proof-of-concept. Its core value lies in highlighting the potential of these four hub genes as a biomarker combination for further validation in future large-scale studies, rather than as a mature diagnostic tool.

4.It is unclear from either the methods or results sections where and how the samples examined in Figure 10 through qRT-PCR were acquired and isolated. This is critical information, please include it.

We appreciate the reviewer's important suggestion. We have supplemented the “Methods” section of the paper with a detailed description of the sample acquisition and isolation procedures used for qRT-PCR analysis in Figure 10. Specifically, we have added information on sample sources, specific isolation methods, and relevant experimental details. This addition of critical information ensures the reproducibility of the research process.

5. Relatedly, given that you are able to perform gene expression analysis, I would very much like to see immunoblot analysis as well. RIPK3 may be upregulated, but whether it’s being cleaved or not is a critical question in regards to its activity, which can be visualized by western blot.

We sincerely appreciate the reviewer's insightful suggestion. We fully concur with your perspective that, beyond gene expression analysis, validation at the protein level—particularly the cleaved activation state of RIPK3—is crucial for confirming its functional role in the cellular programmed necrosis pathway. Western blotting provides a direct visual confirmation of whether RIPK3 undergoes cleavage and its activation status, which cannot be replaced by mRNA-level analysis. We sincerely explain that incorporating this critical validation into the current revision was not feasible due to the constraints of our limited timeframe. Acquiring high-quality, protein-blot-suitable intestinal tissue samples requires initiating new animal experiments, involving rigorous and time-consuming steps such as tissue lysis, protein extraction, and quantification. Completing a new animal study cycle within this period while ensuring data reliability was unattainable. We acknowledge this represents a current limitation of our study. To mitigate this shortcoming and enhance the credibility of our gene screening results, we have supplemented this revision with the following additional work: We introduced an independent clinical sample dataset (GSE232246) to validate the expression levels of the identified hub genes, including RIPK3. simultaneously employing weighted gene co-expression network analysis (WGCNA) to confirm the central role of these hub genes in relevant pathophysiological processes from a network association perspective.

Furthermore, we have outlined a clear plan for subsequent research. Western blot analysis targeting key proteins such as RIPK3 and MLKL will serve as the core experimental component of our next manuscript. This aims to directly validate and expand upon our transcriptomic findings at the protein level, thereby providing a more comprehensive understanding of the underlying molecular mechanisms. We sincerely appreciate your valuable feedback, which has provided crucial guidance for refining our research framework.

6. In the end of your discussion, you state “this study has obvious limitations.” Please actually state what these limitations are.

We sincerely appreciate the reviewer's important suggestion. Following the recommendation, we have elaborated several key limitations of this study in the Discussion section, including the constraints of a small sample size, potential limitations of the technical approach, and inconsistencies in Tnfaip3 expression between transcriptional levels and experimental validation. Clearly articulating these shortcomings not only enhances the completeness and rigor of the research presentation but also provides clear guidance for future research directions. We extend our gratitude once again to the reviewers for their invaluable assistance in refining our manuscript.

Minor Comments

1. In the Introduction, you often introduce a concept like so: “Reperfusion phase: ROS trigger…” or “Intestinal barrier disruption: Intestinal villus tip…” This formatting is confusing, and I would suggest maintaining standard sentence construction.

We appreciate the reviewers' important comments. We have systematically revised the introduction, rewriting paragraphs with unclear conceptual expressions—particularly those describing key processes such as the “reperfusion phase” and “intestinal barrier disruption”—using more standardized and precise sentence structures to ensure logical clarity and terminological consistency.

2. Several points in the methods (e.g. line 100, lines 139-141, etc) are written as instructions rather than a report of what as performed. Please make sure to edit these, for clarity.

We appreciate the reviewer's identification of this critical issue. We have thoroughly reviewed the Methods section (including lines 100, 139–141, etc.) and revised all imperative statements to the indicative past tense required in standard academic writing to accurately describe the operations actually performed. This revision aims to ensure the experimental workflow is presented clearly and accurately, meeting the requirements for reproducibility.

3. Are the 4 control samples in the validation set the same as the 4 controls in the training set? I would hope not, but if so you definitely need new data for the validation set.

We appreciate the reviewer raising this critical point. The situation you describe is indeed accurate. Due to the limited availability of publicly accessible datasets in the field of intestinal ischemia-reperfusion injury, we were unable to identify suitable independent control samples for the validation set in our initial manuscript. We fully understand and agree that using the same control samples as the training set would severely compromise the reliability of validation. Fortunately, we recently discovered a newly uploaded public dataset that meets the required criteria. Accordingly, we have acquired and utilized entirely new independent control samples for the validation set, re-conducted all validation analyses, and ensured the robustness of our conclusions. The relevant modifications have been updated in the paper.

4. Please clarify how you collected necroptosis and pyroptosis related genes from the NCBI database.

The gene sets associated with necroptosis and pyroptosis involved in this study were obtained through systematic retrieval and organization from the NCBI Gene database. Specifically, we utilized the database's advanced search functionality to retrieve mouse genes using relevant keywords. Building upon this foundation, we rigorously reviewed and filtered the preliminary results according to predefined criteria, retaining only genes with well-defined functions, clear nomenclature, and robust supporting evidence in the existing literature. This approach maximized the accuracy and reliability of the gene sets. Ultimately, we identified 168 necrosis-related genes and 299 pyroptosis-related genes. The detailed methodology for this process is described in the Methods section of the paper.

5.Please include vendor as well as product or catalog number in your methods.

We sincerely appreciate the reviewers' identification of this critical shortcoming. We fully agree that providing complete vendor and product number information is fundamental to ensuring the reproducibility of research methods. To this end, we have thoroughly reviewed and revised the “Methods” section, supplementing the relevant materials with the aforementioned details. We deeply apologize for the omission in the previous version, and these modifications significantly enhance the rigor and completeness of the methodology section.

6. Figure 6C is quite information-dense and visually difficult to parse. Please revise.

We appreciate the reviewer's suggestions. We have thoroughly redesigned Figure 6C by simplifying the layout and optimizing the visual presentation, thereby addressing the issues of information overload and visual clutter. The readability of the revised figure has been significantly improved.

7. I would like at least a little bit of mechanistic hypothesizing as to why you saw the inverse relationship with Tnfaip3 in your qRT-PCR results.

We sincerely appreciate this valuable suggestion from the reviewers. We fully agree that proposing a mechanistic hypothesis is crucial for deepening the research conclusions. To this end, we have elaborated in detail in the Discussion section on the “context-dependent degradation” hypothesis for the inconsistent expression of Tnfaip3, suggesting that its protein product may be rapidly degraded in specific inflammatory microenvironments. Your feedback has prompted our study to move beyond mere phenomenological description and take a significant step forward in exploring underlying mechanisms, substantially enhancing the depth of the paper.

8. Line 58-59: this phrasing “In recent years…” makes it sound like the role of cell death in I/IRI is a new phenomenon in nature. I’d suggest editing to say “In recent years, the role of […] has become apparent.” or similar.

We sincerely appreciate the reviewer's insightful corrections. We fully agree that the original phrasing was indeed imprecise and could have misled readers. We have revised it accordingly. Your feedback has helped us more accurately describe the process of deepening scientific understanding, thereby enhancing the rigor of the paper.

9. There’s no need to write drug names in all-capital letters.

We appreciate the reviewer's attention to this formatting issue. We have reviewed all drug names throughout the manuscript (including the Discussion section) and revised them from all-caps to standard writing format as suggested.

10. There’s no need to capitalize “Mast Cells”

We sincerely appreciate the reviewer's attention to this detail. We have thoroughly reviewed the entire manuscript and uniformly corrected the capitalization of all instances of “mast cells” to ensure consistent terminology and uniformity throughout the text. Thank you very much for helping us refine the quality of our manuscript's details.

11. Line 289: please replace “DC Immature” with “immature dendritic cells”

We appreciate the reviewer's corrections. We have amended the term “DC Immature” to “immature dendritic cells” in accordance with the suggestion.

Reviewer #2

Major Comments:

Experimental Design and qRT-PCR Methodology:

1. The manuscript does not describe how the intestinal I/R injury was induced in mice prior to RNA extraction (see Results lines 258–263, Methods section before line 170).The procedure for ischemia induction (artery clamped, duration, reperfusion period, anesthesia, euthanasia) must be clearly detailed to ensure clarity as well as reproducibility. The lack of this information prevents readers from understanding the biological context of the “I/R samples” used for qPCR.

We sincerely appreciate the reviewers' rigorous suggestions. We deeply apologize for the incomplete description of animal experiments in the previous version. Following the feedback, we have detailed th

---

## [Decision Letter · Decision Letter 1]

23 Nov 2025

Comprehensive analysis of genes associated with necroptosis and pyroptosis in intestinal ischemia-reperfusion injury

PONE-D-25-48927R1

Dear Dr. Li,

We’re pleased to inform you that your manuscript has been judged scientifically suitable for publication and will be formally accepted for publication once it meets all outstanding technical requirements.

Kind regards,

Tomasz W. Kaminski

Academic Editor

PLOS ONE

Reviewers' comments:

Reviewer's Responses to Questions

**Comments to the Author**

Reviewer #1: All comments have been addressed

Reviewer #2: (No Response)

2. Is the manuscript technically sound, and do the data support the conclusions?

Reviewer #1: Yes

Reviewer #2: (No Response)

3. Has the statistical analysis been performed appropriately and rigorously?

Reviewer #1: Yes

Reviewer #2: (No Response)

4. Have the authors made all data underlying the findings in their manuscript fully available?

Reviewer #1: Yes

Reviewer #2: (No Response)

5. Is the manuscript presented in an intelligible fashion and written in standard English?

Reviewer #1: Yes

Reviewer #2: (No Response)

Reviewer #1: I would like to offer my sincere thanks to the authors for their comprehensive and thoughtful responses to my inquiries and suggestions. While some differences of opinion may remain, the publication of their data is well-deserved, and I look forward to seeing what fruit their research next bears.

Reviewer #2: (No Response)

**Do you want your identity to be public for this peer review?** For information about this choice, including consent withdrawal, please see our Privacy Policy

Reviewer #1: **Yes: ** Tzvi Pollock

Reviewer #2: No

---

## [Editor Report · Acceptance letter]

PONE-D-25-48927R1

PLOS ONE

Dear Dr. Li,

I'm pleased to inform you that your manuscript has been deemed suitable for publication in PLOS ONE. Congratulations! Your manuscript is now being handed over to our production team.

Kind regards,

on behalf of

Dr. Tomasz W. Kaminski

Academic Editor

PLOS ONE